# Coupling to short linear motifs creates versatile PME-1 activities in PP2A holoenzyme demethylation and inhibition

Yitong Li[1†], Vijaya Kumar Balakrishnan[1], Michael Rowse[1‡], Cheng-Guo Wu[1,2], Anastasia Phoebe Bravos[1], Vikash K Yadav[3], Ylva Ivarsson[3], Stefan Strack[4], Irina V Novikova[5], Yongna Xing[1,2]*

[1]McArdle Laboratory for Cancer Research, Department of Oncology, University of Wisconsin at Madison, School of Medicine and Public Health, Madison, United States; [2]Biophysics program, University of Wisconsin at Madison, Madison, United States; [3]Department of Chemistry – BMC, Uppsala University, Uppsala, Sweden; [4]Department of Neuroscience and Pharmacology, University of Iowa, Iowa City, United States; [5]Environmental Molecular Sciences Laboratory, Pacific Northwest National Laboratory, Richland, United States

*For correspondence:
xing@oncology.wisc.edu

Present address: [†]SRI International Biosciences Division, Harrisonburg, United States; [‡]Indiana University-Purdue University Columbus, Columbus, United States

Competing interest: The authors declare that no competing interests exist.

**Abstract** Protein phosphatase 2A (PP2A) holoenzymes target broad substrates by recognizing short motifs via regulatory subunits. PP2A methylesterase 1 (PME-1) is a cancer-promoting enzyme and undergoes methylesterase activation upon binding to the PP2A core enzyme. Here, we showed that PME-1 readily demethylates different families of PP2A holoenzymes and blocks substrate recognition in vitro. The high-resolution cryoelectron microscopy structure of a PP2A-B56 holo-enzyme–PME-1 complex reveals that PME-1 disordered regions, including a substrate-mimicking motif, tether to the B56 regulatory subunit at remote sites. They occupy the holoenzyme substrate-binding groove and allow large structural shifts in both holoenzyme and PME-1 to enable multipartite contacts at structured cores to activate the methylesterase. B56 interface mutations selectively block PME-1 activity toward PP2A-B56 holoenzymes and affect the methylation of a fraction of total cellular PP2A. The B56 interface mutations allow us to uncover B56-specific PME-1 functions in p53 signaling. Our studies reveal multiple mechanisms of PME-1 in suppressing holoenzyme functions and versatile PME-1 activities derived from coupling substrate-mimicking motifs to dynamic structured cores.

## Editor's evaluation

Xing and colleagues used a combination of biochemical assays and cryo-EM to investigate the role of PME-1 in regulating PP2A, which plays an important role in tumorigenesis. Notably, they reveal here that PME-1 inserts an unstructured loop into the B-domain of the PPA2 holoenzyme and allosterically regulates the activity of the catalytic domain. This novel mechanism then plays a key role in controlling the cellular homeostasis of PP2A. Together, the work presented here provides new insights into mechanisms for an oncogenic function of PME-1 in regulating (inhibiting) p53 phosphorylation via PP2A-B56 holoenzymes under normal and DNA damage response conditions.

## Introduction

Protein phosphatase 2A (PP2A) is a major and the most abundant serine/threonine phosphatase in mammalian cells and dephosphorylates ~half cellular proteins (*Arnold and Sears, 2008*; *Virshup,*

*2000*; *Wlodarchak and Xing, 2016*). The cellular functions of PP2A rely on the formation of structurally distinct diverse heterotrimeric holoenzymes; each consists of a dimeric core enzyme formed by the scaffolding A and catalytic C (C or PP2Ac) subunits and a diverse set of regulatory B subunits. The regulatory subunits belong to four major families (B/B55/PR55, B′/B56/PR61, B″/PR72, and B‴/Striatin), and dictate specific substrate recognition. Recent studies arrive at a merging theme that PP2A holoenzymes recognize substrates by binding to short linear motifs (SLiMs) in substrates (*Hertz et al., 2016*; *Wang et al., 2016b*; *Wang et al., 2016a*; *Wu et al., 2017a*). SLiMs are extremely powerful in synthesizing regulation nodes and signaling networks (*Davey et al., 2015*; *Davey and Morgan, 2016*). Similar observations were also made to substrates of other phosphatases in the PPP family (*Heroes et al., 2013*; *Nasa et al., 2018*; *Ueki et al., 2019*; *Wigington et al., 2020*). PP2A holoenzymes are highly regulated by carboxymethylation of the PP2Ac tail, which is reversibly controlled by PP2A-specific leucine carboxyl methyltransferase (LCMT-1) (*De Baere et al., 1999*; *Lee and Stock, 1993*) and methylesterase (PME-1) (*Lee et al., 1996*). Dysregulation of PP2A holoenzymes or alteration of holoenzyme functions causes many human diseases, including cancer, heart diseases, and neurodegenerative disorders (*Janssens et al., 2005*; *Sontag et al., 2004*). Consistently, LCMT-1 is essential for cell survival (*Lee and Pallas, 2007*), and the deletion of PME-1 in mice is perinatally lethal (*Ortega-Gutiérrez et al., 2008*).

The activity of LCMT-1 and PME-1 is strictly controlled for proper PP2A holoenzyme homeostasis by multiple layers of structural mechanisms. Their activity on the PP2Ac tail, 'TPDYFL' relies on binding to the phosphatase active site, which induces significant changes in LCMT-1 to accommodate the tail (*Stanevich et al., 2011*) and global conformational changes in PME-1 to create the substrate-binding pocket and the active configuration of the catalytic triads (*Xing et al., 2008*). The LCMT-1 activity is enhanced by the activation of PP2Ac and its association with the A-subunit and is thus highly selective toward the active core enzyme, (*Stanevich et al., 2011*; *Stanevich et al., 2014*), ensuring that PP2A is either in latent, inactive or in active substrate-specific forms.

Whether PME-1 also specifically acts on the core enzyme remain controversial. PP2A holoenzymes are predominately methylated in cells (*Tolstykh et al., 2000*; *Wu et al., 2000*; *Yu et al., 2001*), suggesting that PME-1 might not directly demethylate holoenzymes. Holoenzymes are also thermaldynamically the most stable form of PP2A and have nanomolar intersubunit binding affinities (*Chen et al., 2007*). In contrast, PP2A interacts with dynamic regulatory proteins with micromolar binding affinities, such as LCMT-1, PME-1, TIPRL (TOR signaling pathway regulator-like), and α4. Structural overlay of PP2A holoenzymes (*Wlodarchak et al., 2013*; *Xu et al., 2008*; *Xu et al., 2006*) and the PP2A core enzyme–PME-1 complex (*Xing et al., 2008*) indicates that PP2A regulatory subunits would hinder PME-1 binding and prevent it from acting on holoenzymes (*Figure 1a*). A recent discovery, however, starts to question the static view on cellular PP2A holoenzymes. TIPRL/α4 readily disassemble unmethylated PP2A holoenzymes by perturbing the phosphatase active site and inducing global conformational changes in PP2Ac (*Jiang et al., 2013*; *Wu et al., 2017b*). Furthermore, PP2A holoenzymes were found to be cofractionated with PME-1 in tissue extracts (*Longin et al., 2004*; *Tolstykh et al., 2000*). Robust PP2A demethylation was detected after the mammalian cells were lysed (*Yabe et al., 2018*), suggesting that PP2A holoenzymes might be demethylated upon their spatial separation from PME-1 was disrupted.

Could PME-1 directly demethylate PP2A holoenzymes? Here, we demonstrate the ability of PME-1 to interact with and demethylate PP2A holoenzymes from three different families in vitro and the role of the PME-1 disordered regions and holoenzyme-specific substrate-mimicking SLiMs in holoenzyme interactions. The high-resolution cryoelectron microscopy (cryo-EM) structure of a PP2A-B56 holoenzyme–PME-1 complex reveals that PME-1 disordered motifs tether to the holoenzyme, block the substrate-binding groove, and allow large structural shifts in both holoenzyme and PME-1 to accommodate multipartite interactions in the structured cores that are required for methylesterase activation. In addition, PME-1 inhibitor and PME-1–B56 interface mutations allow us to dissect novel PME-1 activities toward p53 signaling. Our studies provide a foundation to investigate the function of dynamic cellular PME-1–holoenzyme interactions.

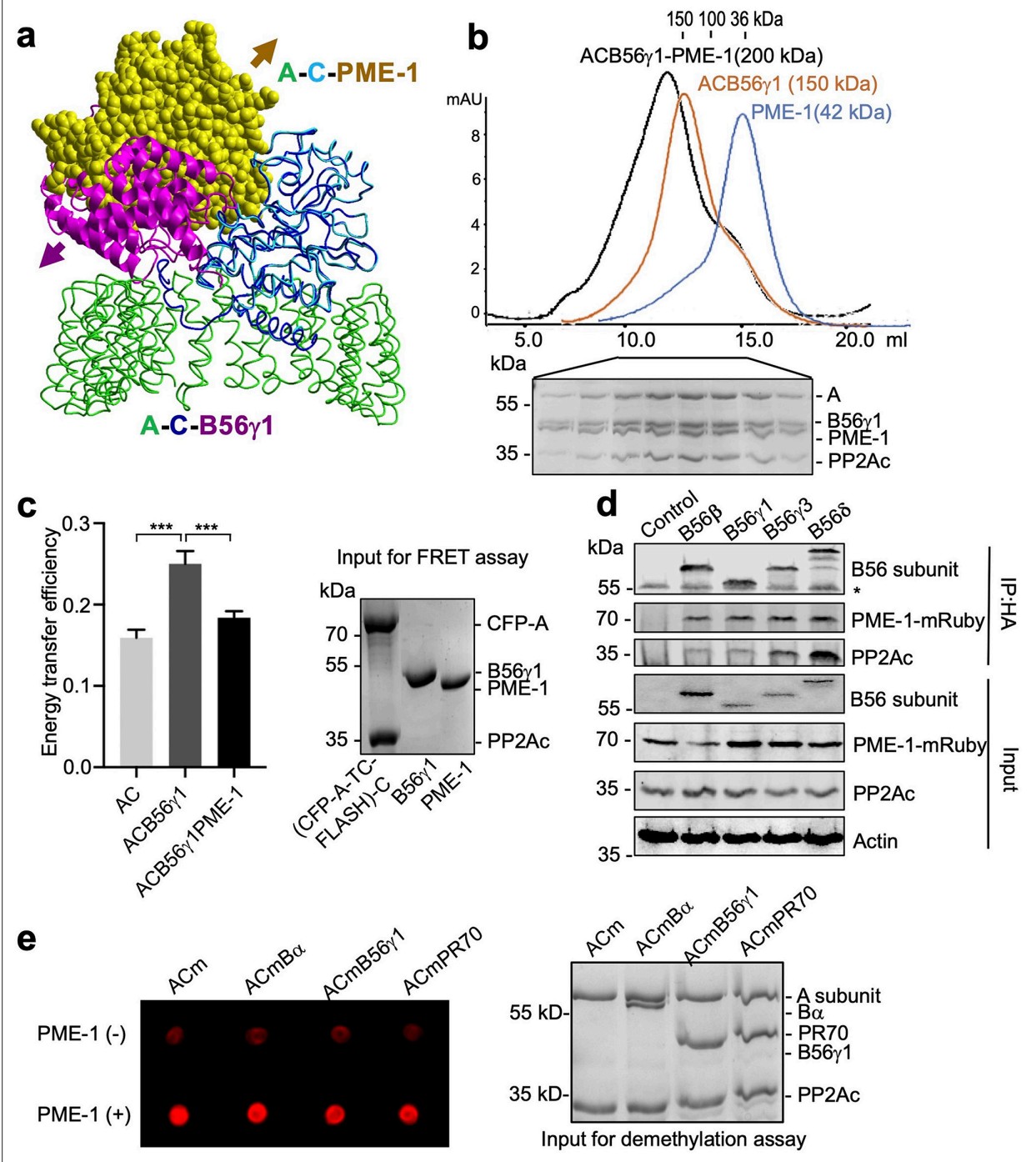

**Figure 1.** PP2A methylesterase 1 (PME-1) directly interacts with and demethylates protein phosphatase 2A (PP2A) holoenzymes. (**a**) Structural overlay of the PP2A-B56γ1 holoenzyme (PDB code: 2NPP) to the PP2A core enzyme–PME-1 complex (PDB code: 3C5W) aligned via PP2Ac (**c**) and the C-terminal five huntingtin-elongation-A-subunit-TOR (HEAT) repeats of the A-subunit. Arrows indicate the directions of movements of PME-1 and B56γ1 needed to avoid clashes in the overlaid structures. (**b**) The overlaid gel filtration profiles of the PP2A-B56γ1 holoenzyme with PME-1 (black), PP2A core enzyme (AC) with B56γ1(orange), and PME-1 alone (blue). The sodium dodecyl sulfate–polyacrylamide gel electrophoresis (SDS–PAGE) data examining protein fractions for the PP2A-B56γ1 holoenzyme with PME-1 were provided below. The molecular weight standards for gel filtration chromatography were generated using the PP2A free catalytic subunit (36 kDa), the PP2A core enzyme (100 kDa), and the PP2A-B56γ1 holoenzyme (150 kDa). (**c**) Fluorescence resonance energy transfer (FRET) assay measured changes in the distance between the A-subunit N- and C-termini in the PP2A core enzyme before and after the addition of B56γ1 with and without PME-1 (left). Representative results were shown with mean ± standard deviation (SD) calculated from three experimental repeats. One-way analysis of variance (ANOVA) with Tukey's multiple comparisons was used to determine the difference between independent groups (***p < 0.001). Protein inputs used in FRET assay were examined by SDS–PAGE and visualized by Coomassie blue staining (right).

*Figure 1 continued on next page*

*Figure 1 continued*

(**d**) Co-immunoprecipitation (co-IP) of PME-1-mRuby and PP2Ac by HA-tagged B56 (B56β, B56γ1, B56γ3, and B56δ) recombinantly expressed in HEK 293T cells. The band with * is the heavy chain of anti-HA antibody. (**e**) PME-1 catalyzes demethylation of methylated core enzyme and holoenzymes in vitro. The level of demethylation was determined by 4b7 antibody that specifically recognizes the unmethylated PP2Ac (left). Inputs of PP2A complexes were examined as in **c** (right).

The online version of this article includes the following source data and figure supplement(s) for figure 1:

**Source data 1.** Source data for *Figure 1b*.

**Source data 2.** Source data for *Figure 1c*.

**Source data 3.** Source data for *Figure 1d*.

**Source data 4.** Source data for *Figure 1e*.

**Figure supplement 1.** PP2A methylesterase 1 (PME-1) interactions with different B56 family members, likely via the conserved B56 common core.

**Figure supplement 1—source data 1.** Source data for *Figure 2—figure supplement 1b*.

**Figure supplement 1—source data 2.** Source data for *Figure 2—figure supplement 1c*.

**Figure supplement 1—source data 3.** Source data for *Figure 2—figure supplement 1d*.

## Results

### PME-1 interacts with PP2A-B56 holoenzymes and induces a more open holoenzyme conformation

Alignment of crystal structures of PP2A holoenzymes and the PP2A core enzyme–PME-1 complex shows that PP2A regulatory subunits exert a huge steric hindrance that would exclude PME-1 binding (*Figure 1a*, *PP2A-B56γ1 holoenzyme shown*). Contradictory to this structural prediction, PME-1 comigrates stoichiometrically with the PP2A-B56γ1 holoenzyme over gel filtration chromatography (*Figure 1b* and *Figure 1—figure supplement 1b*). B56γ1 represents the conserved common core in B56 subunits (*Figure 1—figure supplement 1a*). Consistent with this notion, we also observed stoichiometric comigration of PME-1 and the PP2A-B56ε holoenzyme (*Figure 1—figure supplement 1c*). We probed the holoenzyme conformation using an A-subunit FRET (fluorescence resonance energy transfer) sensor (*Wlodarchak et al., 2013*), in which TC-FLASH produced by tetracysteine peptide (TC) fused to the C-terminus serves as an acceptor for the CFP (cyan fluorescent protein) fused to the N-terminus. The energy transfer efficiency of the PP2A-B56γ1 holoenzyme containing this FRET sensor is much higher than the core enzyme, but is reduced significantly upon the addition of PME-1 (*Figure 1c*). These data showed that that PME-1 interacts with B56 holoenzymes and induces a more open holoenzyme conformation.

We next validated the PME-1–B56 holoenzyme interactions in mammalian cell lysates. We coexpressed PME-1-mRuby fusion protein and recombinant B56 regulatory subunits harboring HA-tag and assessed their interactions by co-immunoprecipitation (co-IP). PME-1-mRuby readily interacts with PP2Ac and multiple HA-tagged regulatory subunits in the B56 family, B56β, B56γ1, B56γ3, and B56δ (*Figure 1d*), supporting the earlier notion that PME-1 interacts with the common core in B56 subunits. Compared to other B56 family members, B56δ interacts with a much higher ratio of PP2Ac over PME-1 (*Figure 1d*), suggesting that the interaction between PME-1 and the PP2A-B56δ holoenzyme might be reduced by other structural elements unique to B56δ (*Figure 1—figure supplement 1a*). Consistently, a much lower stoichiometric amount of PME-1 comigrates with the PP2A-B56δ holoenzyme over gel filtration chromatography (*Figure 1—figure supplement 1d*). The recombinant B56δ associates with PP2Ac at a much higher level compared to other B56 subunits (*Figure 1d*). The resistance of this holoenzyme to PME-1 likely renders it less prone to demethylation and TIPRL/α4-mediated holoenzyme disassembly.

### PME-1 catalyzes direct demethylation of three families of PP2A holoenzymes

PME-1 activation relies on its binding to the PP2A active site (*Xing et al., 2008*). The holoenzyme conformational changes induced by PME-1 above might alleviate steric hindrance and allow PME-1 to interact with the PP2A active site. To test this hypothesis, we assembled the core enzyme and three representative PP2A holoenzymes from three families using Bα, B56γ1, and PR70 with higher

than 90% in vitro methylation. After incubation with PME-1, the methylation level of all three holoenzymes decreases significantly, comparable to that of the core enzyme (*Figure 1e*). Our results suggest that PME-1-PP2A holoenzyme interactions enable all sequential events needed for PME-1 activation, allowing PME-1 to demethylate PP2A holoenzymes directly.

## Mapping of PME-1 disordered regions in binding to PP2A regulatory subunits and holoenzymes

The ability of PME-1 to induce conformational changes in PP2A holoenzymes and overcome the steric hindrance of regulatory subunits led us to search for additional contacts made by PME-1 prior to its interaction with the PP2A active site. PME-1 has two ~40-residue disordered regions at the N-terminus and the internal loop (*Figure 2a*). The latter harbors a SLiM ($^{251}$VEGIIEE$^{258}$E) highly similar to B56 substrates that interact with the conserved B56 common core via a signature motif 'LxxIxE' (*Hertz et al., 2016*; *Wang et al., 2016b*; *Wang et al., 2016a*; *Wu et al., 2017a*). The GST-tagged PME-1 B56 SLiM peptide (247–263) binds to B56γ1 with a binding affinity of ~13 µM measured by isothermal titration calorimetry (ITC). This observation intrigued us to map the function of the disordered regions of PME-1 in their interactions with PP2A regulatory subunits (*Figure 2a*).

Full-length PME-1 (PME-1 FL) interacts with B56γ1 with a binding affinity of 0.4 µM (*Figure 2— figure supplement 1a*, *left*). The deletion of the internal loop harboring the B56-SLiM (ΔIL) abolishes their interaction completely and deletion of N-terminal 18 residues (ΔN18) reduces the binding affinity by ~eightfold (*Figure 2—figure supplement 1a*, *middle and right*). We observed that a complete deletion of the N-terminal disordered region or deletion of N-terminal residues beyond residue 18 tends to cause PME-1 aggregate or a poorer protein behavior. PME-1 ΔN18 was thus used to test the function of the N-terminal disordered region. We also mapped the contribution of PME-1 motifs in interactions with PR70 and Bα. PME-1 binds directly to PR70 with a binding affinity of 0.7 µM. ΔIL reduces the binding affinity by ~twofold, and ΔN18 completely abolishes this interaction (*Figure 2— figure supplement 1b*). Using titration pulldown assay, we showed that PME-1 also interacts with Bα, which was weakened by ΔN18 but barely affected by ΔIL (*Figure 2—figure supplement 1c*). We did not use ITC for Bα due to its relatively lower thermal stability. As summarized in *Figure 2a*, these data indicate that the PME-1 internal loop harbors a primary B56-docking motif, and the N-terminal disordered region possesses a Bα- and PR70-docking motif.

We further investigated how PME-1 disordered regions contribute to the interactions with holoenzymes. PME-1 comigrates with all three PP2A holoenzymes over gel filtration chromatography (*Figure 2b–d*). ΔN18 decreased PME-1 interactions with all three PP2A holoenzymes (*Figure 2b–d*), consistent with the critical role of the N-terminal motif in binding to all three regulatory subunits (*Figure 2a*). Similarly, ΔIL had a major effect on interaction with the PP2A-B56γ1 holoenzyme (*Figure 2b*) but a minor effect on interactions with PP2A-Bα and PP2A-PR70 holoenzymes (*Figure 2c, d*).

Collectively, the data summarized in *Figure 2a* demonstrate that the N-terminal motif of PME-1 contributes to the interaction with all three regulatory subunits and holoenzymes with an essential role for interactions with Bα and PR70. The internal loop plays a dominant role in interaction with B56 but a minor role for interactions with Bα and PR70. Despite the dual interactions of the PME-1 core to the PP2A active site and PP2Ac-tail (*Xing et al., 2008*), these data demonstrate direct interactions between PME-1 disordered motifs and PP2A regulatory subunits that are essential for interactions with PP2A holoenzymes.

## PME-1-binding inhibits substrate recognition by PP2A holoenzymes

Since the B56-binding motif in the PME-1 internal loop is similar to SLiMs in substrates (*Figure 2a*), we examined the potential role of PME-1 in blocking substrate recognition by PP2A holoenzymes. Previously, we elucidated the SLiM-based interactions that target PP2A-B56 to its substrates using proteomic peptide phage display (ProP-PD). SYT16 peptide was the strongest hit of B56 substrates identified by ProP-PD (*Wu et al., 2017a*). The GST-SYT16 peptide (LSSIAEEE) binds to B56γ1 with a binding affinity of 0.6 µM measured by ITC. By competition pulldown assay, we showed that the interaction between the SYT16 peptide and PP2A-B56γ1 was blocked by increasing concentrations of PME-1 (*Figure 2e*, left and *Figure 2—figure supplement 3a*). No competitions were detected

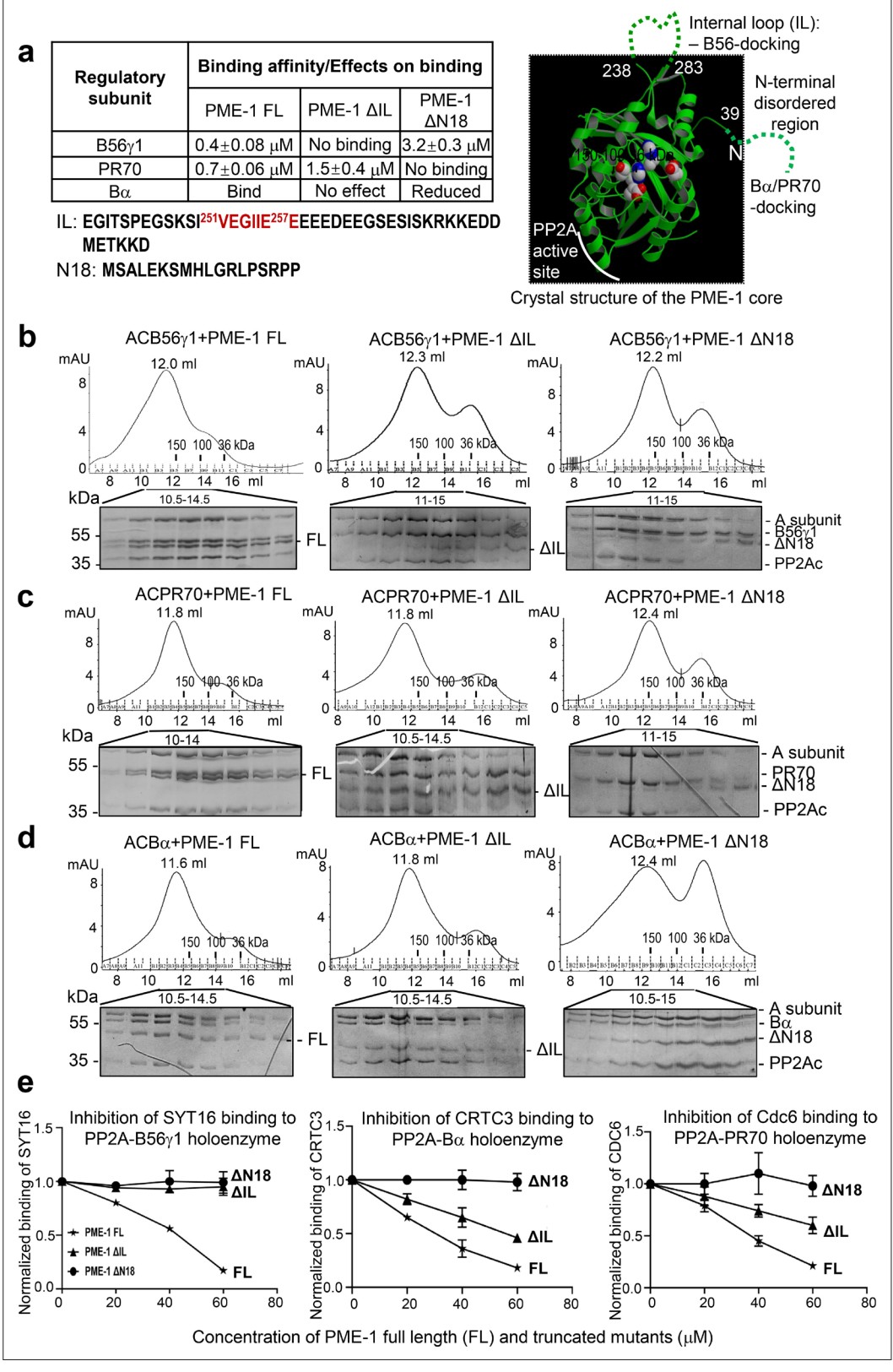

**Figure 2.** Mapping of PP2A methylesterase 1 (PME-1) interactions with protein phosphatase 2A (PP2A) regulatory subunits and holoenzymes. (**a**) Summary of mapping results (*Figure 2—figure supplement 1*) on the roles of PME-1 disordered regions in interactions with different PP2A regulatory subunits (left) and illustration of disordered regions (dashed lines) and their contributions on the crystal structure of the apo-PME-1 structured core (PDB code:

*Figure 2 continued on next page*

*Figure 2 continued*

3C5V) (right). Sequences of PME-1 internal loop (IL) and N-terminal 18 residues (N18) were shown, highlighting a substrate-mimicking B56 short linear motif (SLiM) in IL (lower left). The boundary residue numbers for the disordered regions are labeled, and the PME-1 active site residues are highlighted in spheres (right). Comigration of PP2A-B56γ1 (**b**), PP2A-PR70 (**c**), or PP2A-Bα (**d**) holoenzymes with PME-1 FL, ΔIL, or ΔN18 over gel filtration chromatography. Protein fractions with the indicated ranges of elution volumes were analyzed by sodium dodecyl sulfate–polyacrylamide gel electrophoresis (SDS–PAGE) and visualized by Coomassie blue staining. The molecular weight standards for gel filtration chromatography are generated as in *Figure 1b*. (**e**) PME-1 concentration-dependent inhibition of substrate peptide binding to specific holoenzymes. Inhibition curves against GST-SYT16 ($^{132}$KLPHVLSSIAEEEHH$^{147}$L) binding to PP2A-B56γ1 (left), GST-CRTC3 ($^{380}$SGPSRRRQPPVSPLTLSPGPE$^{401}$A) binding to PP2A-Bα (middle), and GST-Cdc6 ($^{49}$KALPLSPRKRLG DDNLCNTPHLPPCSPPKQGK KENGPPHSH$^{90}$T) to PP2A-PR70 (right) by PME-1 FL, ΔN18, or ΔIL were generated from competitive pulldown data in *Figure 2—figure supplement 3a*. Values for all data points on the inhibition curves are mean ± standard deviation (SD) from three experimental repeats.

The online version of this article includes the following source data and figure supplement(s) for figure 2:

**Source data 1.** Source data for *Figure 2b*.

**Source data 2.** Source data for *Figure 2c*.

**Source data 3.** Source data for *Figure 2d*.

**Source data 4.** Source data for *Figure 2e*.

**Figure supplement 1.** PP2A methylesterase 1 (PME-1) interactions with Bα and PR70 regulatory subunits and holoenzymes.

**Figure supplement 1—source data 1.** Source data for *Figure 2—figure supplement 1c*.

**Figure supplement 2.** Mapping and characterization of CRTC3 peptide motif that interacts with Bα regulatory subunit.

**Figure supplement 2—source data 1.** Source data for *Figure 2—figure supplement 2b*.

**Figure supplement 2—source data 2.** Source data for *Figure 2—figure supplement 2c*.

**Figure supplement 3.** PP2A methylesterase 1 (PME-1) inhibits substrate binding to protein phosphatase 2A (PP2A) holoenzymes.

**Figure supplement 3—source data 1.** Source data for *Figure 2—figure supplement 3a*.

---

between PME-1 ΔIL or PME-1 ΔN18 and this PP2A-B56γ1 substrate peptide (*Figure 2e*, left and *Figure 2—figure supplement 3a*).

No consensus motifs have been published for the substrates of PP2A-Bα and PP2A-PR70 holo-enzymes. Nonetheless, peptide motifs from a few substrates have been identified or are under investigation for these holoenzymes. Bα was recently reported to interact with CREB-regulated tran-scription coactivator 3 (CRTC3) and regulates CRTC3 phosphorylation and CREB transcriptional activity (*Sonntag et al., 2019*). To investigate the effect of PME-1 on substrate recognition by the PP2A-Bα holoenzyme, we mapped the disordered regions of CRTC3, and identified a peptide motif that specif-ically recognizes this holoenzyme (*Figure 2—figure supplement 2a–c*). PME-1 competes with this CRTC3 motif for interaction with the PP2A-Bα holoenzyme (*Figure 2e*, middle and *Figure 2—figure supplement 3a*). The PR70 regulatory subunit was previously shown to recognize a peptide motif in Cdc6 and control the cellular level of Cdc6 during cell cycle (*Davis et al., 2008*; *Wlodarchak et al., 2013*). Similarly, PME-1 competes with Cdc6 for binding to the PP2A-PR70 holoenzyme (*Figure 2e*, right and *Figure 2—figure supplement 3a*).

Consistent with the critical role of the PME-1's N-terminal motif in holoenzyme interactions, no competition was detected between PME-1 ΔN18 and the substrates of all three holoenzymes (*Figure 2e* and *Figure 2—figure supplement 3a*). PME-1 ΔIL completely lost the ability to block substrate recognition by the PP2A-B56γ1 holoenzyme, but not the PP2A-Bα and PP2A-PR70 holoen-zymes (*Figure 2e* and *Figure 2—figure supplement 3a*), consistent with the critical role of this loop for interactions with PP2A-B56γ1, but not the other two holoenzymes. Parallel to ProP-PD phage selection to identify SLiMs for the PP2A-B56γ1 holoenzyme (*Wu et al., 2017a*), we also performed ProP-PD for the holoenzyme in complex with PME-1. The data showed that the presence of PME-1 blocked the binding of all peptide motifs recognized by the PP2A-B56γ1 holoenzyme and reduced the counts for all B56-binding motifs to zero in phage selection (*Figure 2—figure supplement 3b*).

Our data demonstrate that PME-1 binding can block substrate recognition of holoenzymes from three families, likely by PME-1's substrate-mimicking holoenzyme-docking motifs (*Figure 2—figure supplement 3c*).

## Overall structure of the PP2A-B56γ1–PME-1 complex

To define the structural and molecular basis of PME-1 interaction with PP2A holoenzymes and dissect its multifaceted activities, we determined a three-dimensional structure of the PP2A-B56γ1–PME-1 heterotetrameric complex using single-particle cryo-EM. To trap the enzymatic intermediate, we assembled the complex using the fully methylated PP2A-B56γ1 holoenzyme and an inactive PME-1 mutant, S156A (*Xing et al., 2008*), followed by covalent crosslinking using glutaraldehyde. After extensive 2D and 3D classifications and careful separation of the tetrameric complex particles from the unbound holoenzyme, the structure was finally determined at an overall resolution of 3.4 Å (*Figure 3—figure supplements 1–3* and *Table 1*). The PP2A-B56γ1–PME-1 complex adopts a pentagram architecture with a size of 100 × 100 × 90 Å° (*Figure 3a*). The structure reveals multiple B56–PME-1 interaction interfaces, large conformational changes in both the holoenzyme and PME-1, and mechanisms for PME-1's multifaceted activities.

The PME-1-bound holoenzyme has a similar overall architecture to the holoenzyme alone and maintains the majority of intersubunit interfaces, but exhibits three major structural changes. Structures overlaid via PP2Ac showed that the last five huntingtin-elongation-A-subunit-TOR (HEAT) repeats of the A-subunit remain mostly unchanged. The N-terminal 10 HEAT repeats shift significantly by increasing distances of 4–12 Å from HEAT repeat 10 to 1, resulting in a 12 Å shift in B56 that alleviates the steric hindrance for PME-1 binding (*Figure 3b*). This structural observation is consistent with the signal changes in the holoenzyme FRET sensor in response to PME-1 binding (*Figure 1c*). The structural shifts in B56 (*Figure 3b*) allow juxtaposition of the PP2Ac–PME-1 interactions between PME-1 M342/Val 343 and PP2Ac Val126/Y127 to a portion of the PP2Ac–B56 interface in the holoenzyme–PME-1 complex (*Figure 3c*, *left*). The juxtaposed interfaces are associated with minor changes to this portion of the PP2Ac–B56 interface in the holoenzyme (*Figure 3c*, *right*). Most prominently, the interactions of the B56γ1 internal loop (IL, 110–130) to PP2Ac near the phosphatase active site in the holoenzyme (*Figure 3d*, *right*) were abolished in the holoenzyme–PME-1 complex (*Figure 3d*, *left*). The perturbation of this loop might be at least in part due to PME-1 binding to the PP2A active site. Finally, the methylated PP2Ac tail no longer occupies the A–B56 interface compared to the holoenzyme. The latter two observations are described in detail later.

The mode of PME-1 binding underlies four coherent mechanisms that suppress the holoenzyme functions. In addition to binding directly to the phosphatase active site and demethylating the holoenzyme, PME-1 also occupies the B56 protein groove for recruiting substrates. PME-1 undergoes a significant angular movement away from B56 up to 6 Å pivoted at the PME-1 helix that contacts the phosphatase active site (*Figure 3e*). The PME-1-PP2A active site interface remains essentially the same compared to the core enzyme–PME-1 complex (*Figure 3f*). It is important to note that Arg268 in PP2Ac near the phosphatase active site forms several H-bond interactions with PME-1 (*Figure 3f*), displacing its multiple H-bond contacts with the B56 internal loop in the holoenzyme (*Figure 3d*). The partially overlapping interfaces likely prevent the interactions of the B56 internal loop to PP2Ac in the holoenzyme–PME-1 complex (*Figure 3d–f*). The negatively charged B56 internal loop would facilitate the phosphorylation site next to a positive patch to enter the holoenzyme active site (*Figure 3—figure supplement 4*). PME-1 binding and perturbation of the B56 internal loop in the holoenzyme would suppress this mechanism. The PME-1 angular shift further accommodates B56–PME-1 interactions. PME-1 ungergoes global allosteric changes for methylesterase activation as revealed by structural comparison of the PME-1 apo-enzyme and its complex with the PP2A core enzyme (*Xing et al., 2008*). The PME-1's angular structural shift in its complex with the holoenzyme underscores the ability of PME-1 to undergo different modes of dynamic changes.

## B56–PME-1 interfaces

B56γ1 interacts with PME-1 at three separate interfaces involving both PME-1 structured core and disordered regions (*Figure 4a*). Consistent with the mapping and sequence analysis earlier (*Figure 2*), the B56-docking SLiM in PME-1 occupies the substrate-binding groove (*Figure 4a*), similar to the substrate peptide from BubR1 ([669]LDPIIE[675]D) (*Wang et al., 2016b*). Five residues (V251, E252, I254,

**Table 1.** Cryoelectron and 36 are marked as headers microscopy (cryo-EM) data collection, model building, and structure refinements for the protein phosphatase 2A (PP2A)-B56 $\gamma$ 1–PP2A methylesterase 1 (PME-1) complex.

**Summary of data collection and model statistics**

| Data collection and processing | |
| --- | --- |
| Number of grids used | 1 |
| Grid type | Quantifoil 300 mesh R 1.2/1/3 with ultrathin carbon |
| Microscope | Titan Krios |
| Detector | Gatan K3 Summit |
| Voltage (kV) | 300 |
| Electron dose (e−/Å$^2$) | 50.8 |
| Defocus range (µm) | 1.5–2.3 |
| Pixel size (A) | 1.059 |
| Number of movies | 7529 |
| Number of particles | 276,737 |
| PDB | 7SOY |
| EMD | EMD-25363 |
| Symmetry | C1 |
| Map resolution (Å) | 3.4 |
| FSC threshold | 0.143 |
| **Refinement (Phenix)** | |
| Initial model used (PDB code) | 3C5W, 2NPP |
| Resolution (Å) | 3.4 |
| Map CC | 0.84 |
| Map sharpening $B$ factor (Å$^2$) | −10 |
| **Model composition** | |
| Number of chains | 4 |
| Nonhydrogen atoms | 12180 |
| Protein residues/waters | 1529/0 |
| Ligands/metals | 0/0 |
| **R.m.s. deviations** | |
| Bonds length (Å) | 0.003 |
| Bonds angle (°) | 0.572 |
| **Validation** | |
| MolProbity score | 1.66 |
| Clashscore | 13.36 |
| Rotamer outerliers (%) | 0 |
| C-Beta outerliers (%) | 0 |
| **Ramachandran plot statistics (%)** | |
| Favored | 98.42 |
| Allowed | 1.58 |
| Outlier | 0 |

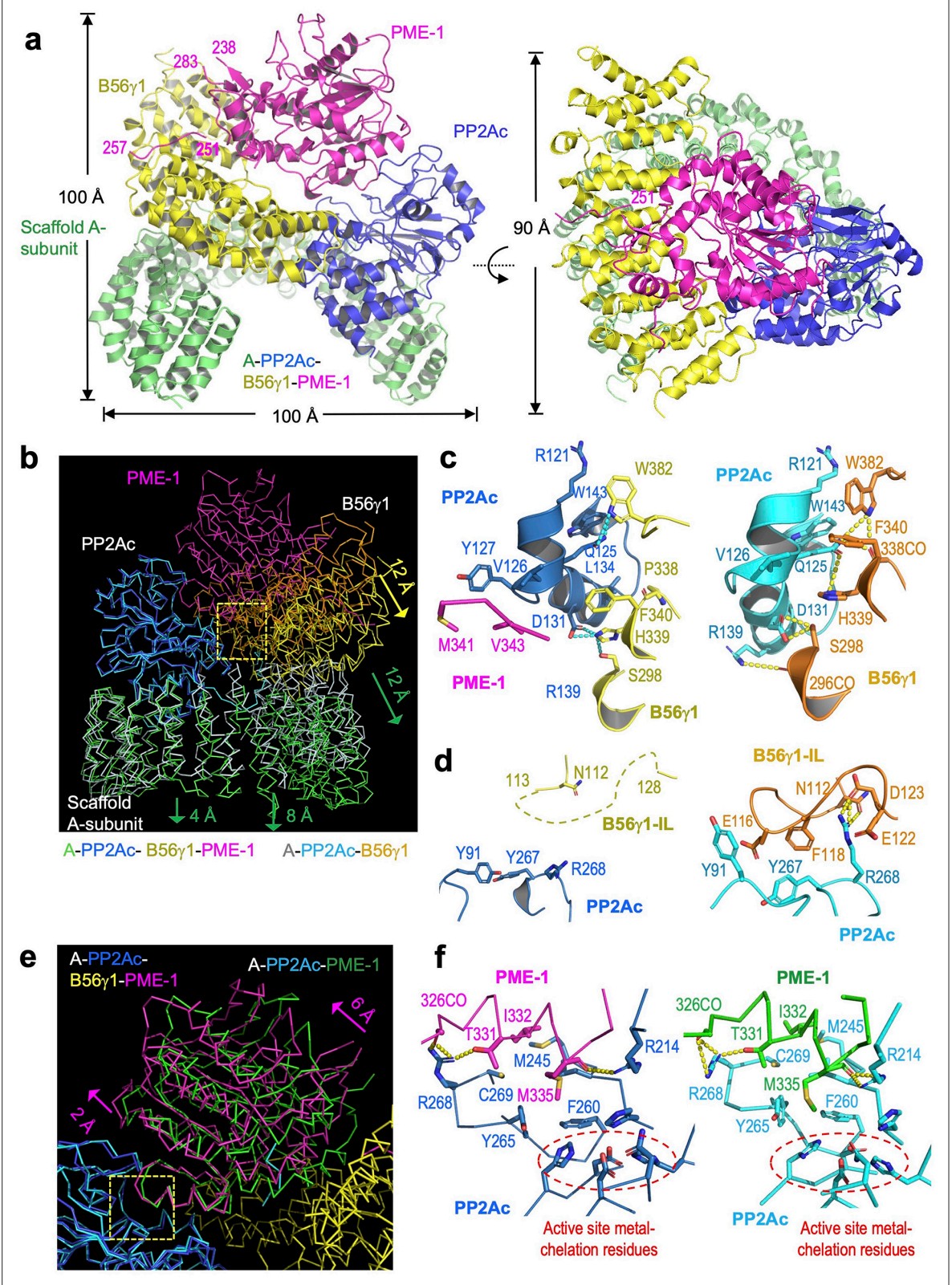

**Figure 3.** Cryoelectron microscopy (cryo-EM) structure of the protein phosphatase 2A (PP2A)-B56γ1–PP2A methylesterase 1 (PME-1) complex. (**a**) Overall structure of the PP2A-B56γ1–PME-1 complex. Two perpendicular views are shown. The A-subunit, PP2Ac, B56γ1, and PME-1 are shown in cartoon and colored green, blue, yellow, and magenta, respectively. The boundary residue numbers of the disorderd PME-1 internal loop (238 and 283) and the visible B56 short linear motif (SLiM) within it (251 and 257) were labeled. (**b**) Structural comparison of the PP2A-B56γ1–PME-1 complex

*Figure 3 continued on next page*

*Figure 3 continued*

and the PP2A-B56γ1 holoenzyme (PDB code: 2NPP), showing the overall structures overlaid via PP2Ac. (**c, d**) Side-by-side closeup views of the B56γ1/ PP2Ac interfaces for the B56γ1 core and internal loop, respectively. (**e**) Structural comparison of the PP2A-B56γ1–PME-1 complex and the PP2A core enzyme–PME-1 complex (PDB code: 3C5W), showing the overall structures overlaid via PP2Ac. (**f**) Side-by-side closeup views of the PP2A active site– PME-1 interfaces. For b–f, models are shown in ribbon for overall structures and cartoon with residues in stick for closeup views. The PP2A-B56γ1–PME-1 complex is colored as in **a**. The A-subunit, PP2Ac, and B56γ1 in the holoenzyme are colored helium, cyan, and orange, respectively. PP2Ac and PME-1 in the PP2A core enzyme–PME-1 complex are colored cyan and green, respectively.

The online version of this article includes the following figure supplement(s) for figure 3:

**Figure supplement 1.** Cryoelectron microscopy (cryo-EM) data processing and model building.

**Figure supplement 2.** Representative cryoelectron microscopy (cryo-EM) density maps for the structural fragments in the A-subunit, B56γ1, PP2Ac, and PP2A methylesterase 1 (PME-1) of the protein phosphatase 2A (PP2A)-B56γ1–PME complex.

**Figure supplement 3.** A minor 3D class gives a low-resolution model for the protein phosphatase 2A (PP2A)-B56γ1 holoenzyme.

**Figure supplement 4.** Illustration of the role of B56 internal loop in facilitating substrate entry to the holoenzyme active site.

I255, and E256) in this SLiM form sidechain interactions with the B56γ1 substrate-binding groove (*Figure 4b*, *interface I*). Next to this protein groove features a network of salt-bridge and H-bond interactions between the PME-1 core (residues N192, Q195, N196, and R199) and B56γ1 (residues D180, K183, E226, and Q266), centered at the R199–E226 salt bridge interaction (*Figure 4c*, *interface II*). Interface III is two HEAT repeats away from interface I and involves both the PME-1 structured core and its N-terminal disordered region (1–40) (*Figure 4a, d*). This interface harbors three widely separated salt bridge interactions between PME-1 residues, K217, R39, and R37, and B56γ1 residues, D313, E353, and E399 (*Figure 4d*). The rest of the N-terminal disordered region (1–36) is invisible in the electron density map, but our earlier mapping of PME-1 disordered regions suggested that PME-1 N-terminal residues 1–18 also contribute to B56γ1 interaction (*Figure 2*), indicating the fourth interface between PME-1 and B56. The mode of PME-1 binding is likely common for all B56 subunits as it predominantly involves the B56 common core (*Figure 1—figure supplement 1a*).

To assess the function of the B56–PME-1 interfaces, we introduced mutations to PME-1 and B56γ1 residues at the above three interfaces. All single or combined interface mutations in either PME-1 or B56γ1 weakened their interactions (*Figure 4e, f*), underlying that all three interfaces are important. Furthermore, PME-1 mutations from each interface also reduced PME-1 binding to the PP2A-B56ε holoenzyme, another B56 family member; the combined mutations from three interfaces (I254K/ R199E/R39E), hereafter referred to as 3MU, completely disrupted this binding (*Figure 4g*). These results demonstrate that B56–PME-1 interfaces are crucial for PME-1 interactions with different PP2A-B56 holoenzymes, consistent with the earlier data that detected interactions between the recombinant PME-1 and multiple B56 subunits in mammalian cell lysates (*Figure 1d*).

## Selective effects of PME-1 mutations at B56 interfaces compared to PME-1 inhibitor

We reason that B56–PME-1 interfaces dictate PME-1's activity specifically toward PP2A-B56 holoenzymes, but not the core enzyme or holoenzymes from other families. Consistent with this notion, 3MU, the combined mutations from three interfaces characterized above (*Figure 4g*), significantly reduced the methylesterase activity toward the PP2A-B56γ1 holoenzyme (*Figure 5a* and *Figure 5—figure supplement 1a*), but not toward the core enzyme, or PP2A-Bα/PP2A-PR70 holoenzymes (*Figure 5a* and *Figure 5—figure supplement 1b–d*). In contrast, ABL127, a compound that blocks PME-1's activity by producing an enzyme-inhibitor adduct at the PME-1 active site (*Bachovchin et al., 2011*), reduced PME-1 binding to all PP2A oligomeric complexes examined, including the core enzyme and the PP2A-Bα/PP2A-B56ε/PP2A-PR70 holoenzymes (*Figure 5b*). These results also indicate that methylesterase activation and the entry of PP2Ac tail into the PME-1 active site are essential for PME-1–holoenzyme interactions. Consistently, the cryo-EM electron density map for the PP2Ac tail is barely retained at the A–B56γ1 interface, sharply different from the holoenzyme (*Figure 5c*). The disappearance of PP2Ac tail from this interface is well defined by the high local resolution and low protein dynamics at this interface (*Figure 5—figure supplement 2*). While the methylesterase activity toward the holoenzyme is detected biochemically, the entry of the PP2Ac tail into the PME-1 active

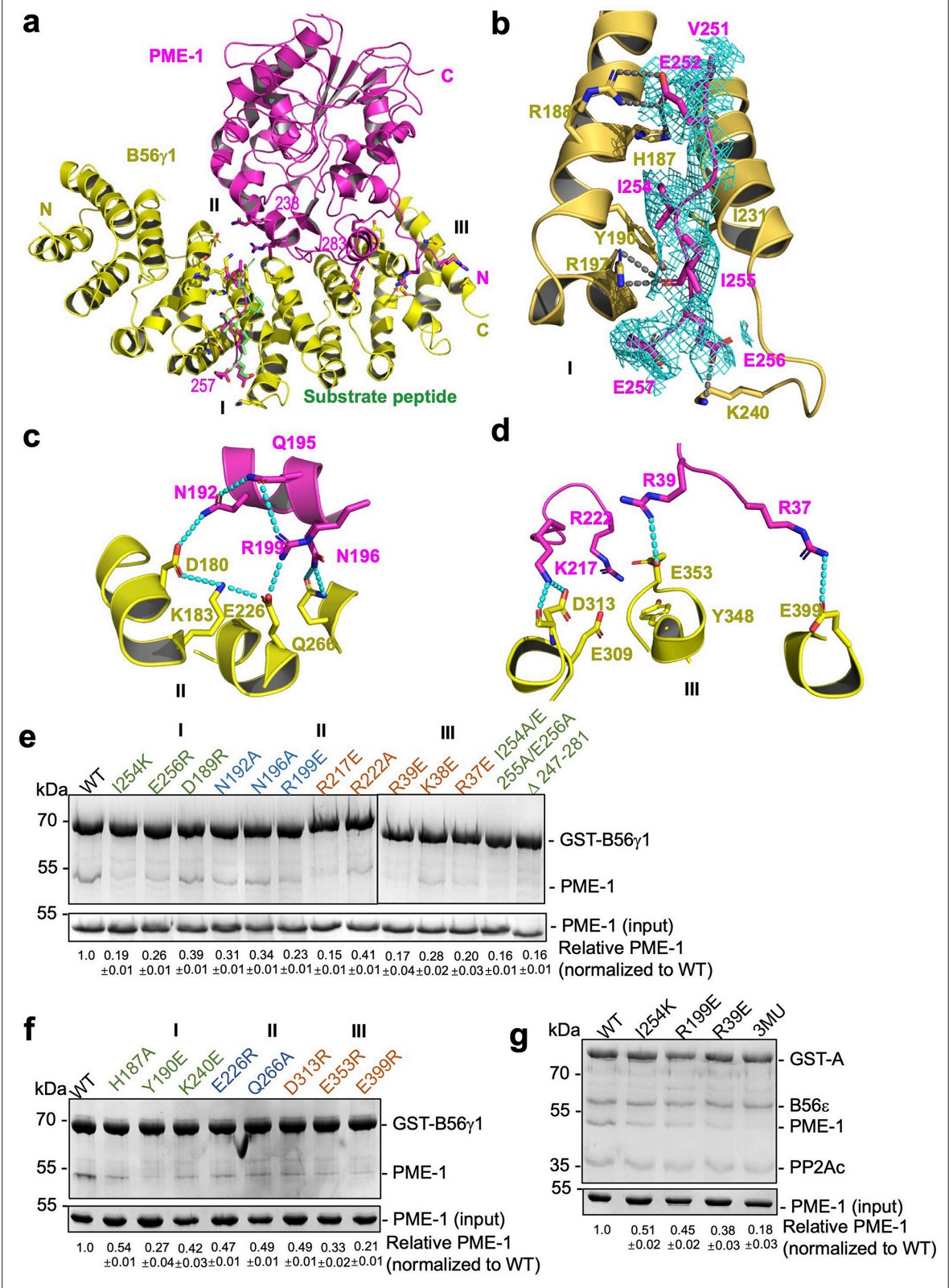

**Figure 4.** B56–PP2A methylesterase 1 (PME-1) interfaces. (**a**) An overview of interactions between B56γ1 (yellow) and PME-1 (magenta). The short linear motif (SLiM) in the PME-1 internal loop was perfectly overlaid with the BubR1 substrate peptide ($^{669}$LDPIIE$^{675}$D) from the B56γ1–BubR1 complex (PBD code: 5JJA) aligned by B56γ1. (**b–d**) Closeup views of interaction interfaces I, II, and III between B56γ1 and PME-1. B56γ1 and PME-1 residues are shown in stick and colored yellow and magenta, respectively. The cryoelectron microscopy (cryo-EM) electron density map for the SLiM in PME-1 was colored

*Figure 4 continued on next page*

*Figure 4 continued*

cyan and shown at 4 Å. (**e**) Pulldown of PME-1 WT or mutants by GST-B56γ1. (**f**) Pulldown of PME-1 by GST-B56γ1 bearing WT or mutant B56γ1. (**g**) Pulldown of PME-1 WT or mutants by the GST-tagged protein phosphatase 2A (PP2A)-B56ε holoenzyme. (**e–g**) GST-tagged proteins were immobilized on Glutathione-Sepharose 4B (GS4B) resins before pulldown. Proteins associated with GS4B resins after pulldown were examined by sodium dodecyl sulfate–polyacrylamide gel electrophoresis (SDS–PAGE) and visualized by Coomassie blue staining. The intensities of bound PME-1 were first normalized to the immobilized GST-tagged proteins and then normalized against the WT (1.0). The quantified results of three independent experiments were shown as mean ± standard deviation (SD) at the bottom of the gel.

The online version of this article includes the following source data for figure 4:

**Source data 1.** Source data for *Figure 4e*.

**Source data 2.** Source data for *Figure 4f*.

**Source data 3.** Source data for *Figure 4g*.

site pocket is not captured by the cryo-EM structure, likely because this pocket is highly dynamic and is the most dynamic structure in the complex (*Figure 5—figure supplement 2*).

Our structural and biochemical observations collectively arrive at a 'latch-to-induce-and-lock' model for PME-1 interaction with holoenzymes and methylesterase activation (*Figure 5d*). Initial latching of PME-1 disordered regions triggers holoenzyme conformational changes, allowing PME-1 to make dual contacts to the PP2Ac active site and tail. These contacts lock a stable interaction and activate the methylesterase activity toward holoenzymes. It is important to mention that the length of the PME-1 internal loop (239–282) is crucial in this 'latch-to-induce-and-lock' model for the B56 SLiM ($^{251}$VEGIIE$^{257}$E) within the loop to reach the substrate SLiM-binding pocket of B56. As reflected in *Figure 3a*, the internal loop provides two invisible 12- and 25-residue linkers, 239–250 and 258–282, to span the direct distances of 23 Å and 31 Å, respectively, from the enzyme core to the two termini of this SLiM. Deleting any residues in the first linker might compromise the 'latch-to-induce-and-lock' model.

## Uncovering and dissecting cellular PME-1 functions in p53 signaling

The selective effects of PME-1 mutations at B56 interfaces on the PP2A-B56 holoenzymes provide us an opportunity to dissect multifaceted PME-1 functions in mammalian cells. In response to DNA damage, p53 induces cell cycle arrest and apoptosis. Previous studies showed that PP2A facilitates p53 activation by targeting pThr55, an inhibitory phosphorylation site that reduces p53 stabilization (*Li et al., 2007*; *Wu et al., 2014*). We next dissected PME-1's functions in p53 signaling using B56 interface mutations and a PME-1 inhibitor, ABL127. Overexpression of PME-1-mRuby elevates p53 phosphorylation at Thr55, and ABL127 reduces this elevation (*Figure 6a, b*). B56 interface mutations, 3MU and ΔIL, abolish this PME-1 activity comparable to or better than ABL127 (*Figure 6a, b*). In contrast, PME-1 3MU and ΔIL affect the cellular PP2A methylation less than ABL127 (*Figure 6a, b*). These data demonstrate a novel role of PME-1 in regulating p53 phosphorylation and pinpoint this cellular function to its activity toward PP2A-B56 holoenzymes. Consistently, p53 pThr55 was previously shown to be a target site of PP2A-B56 holoenzymes (*Li et al., 2007*).

Next, we further demonstrated the role of PME-1 in p53 signaling during DNA damage response (DDR). Upon doxorubicin treatment to induce DDR, both total p53 protein and pThr55 were increased over time and accumulated to high levels after 24 hr in 293T cells (*Figure 6c, d*). During DDR, the presence of ABL127 led to a more rapid p53 accumulation, accompanied by an attenuated increase in pThr55 (*Figure 6c, d*). PME-1–B56 interface mutations gave similar effects during DDR with PME-1-mRuby overexpression. PME-1 3MU led to a more rapid p53 accumulation and attenuated increase in pThr55 compared to WT PME-1 (*Figure 6e, f*). These results underscore an inverse relationship between pThr55 and p53 stability and the role of PME-1 in suppressing p53 accumulation by enhancing pThr55 during DDR. ABL127 and PME-1 3MU reduced the ratio of pThr55 over total p53 in a similar scale during the time course of DDR (*Figure 6g*), suggesting that the role of PME-1 in regulating pThr55 of p53 is primarily mediated by its interactions with PP2A-B56 holoenzymes.

In summary, we demonstrate the role of PME-1 in inhibiting tumor suppressor p53 signaling by suppressing the function of PP2A-B56 holoenzymes toward pThr55 of p53 (*Figure 6h*). These observations corroborate with the oncogenic function of PME-1. PME-1 amplification is found in many types of cancer and is associated with poorer survival outcomes (*Figure 6—figure supplement 1*). The B56

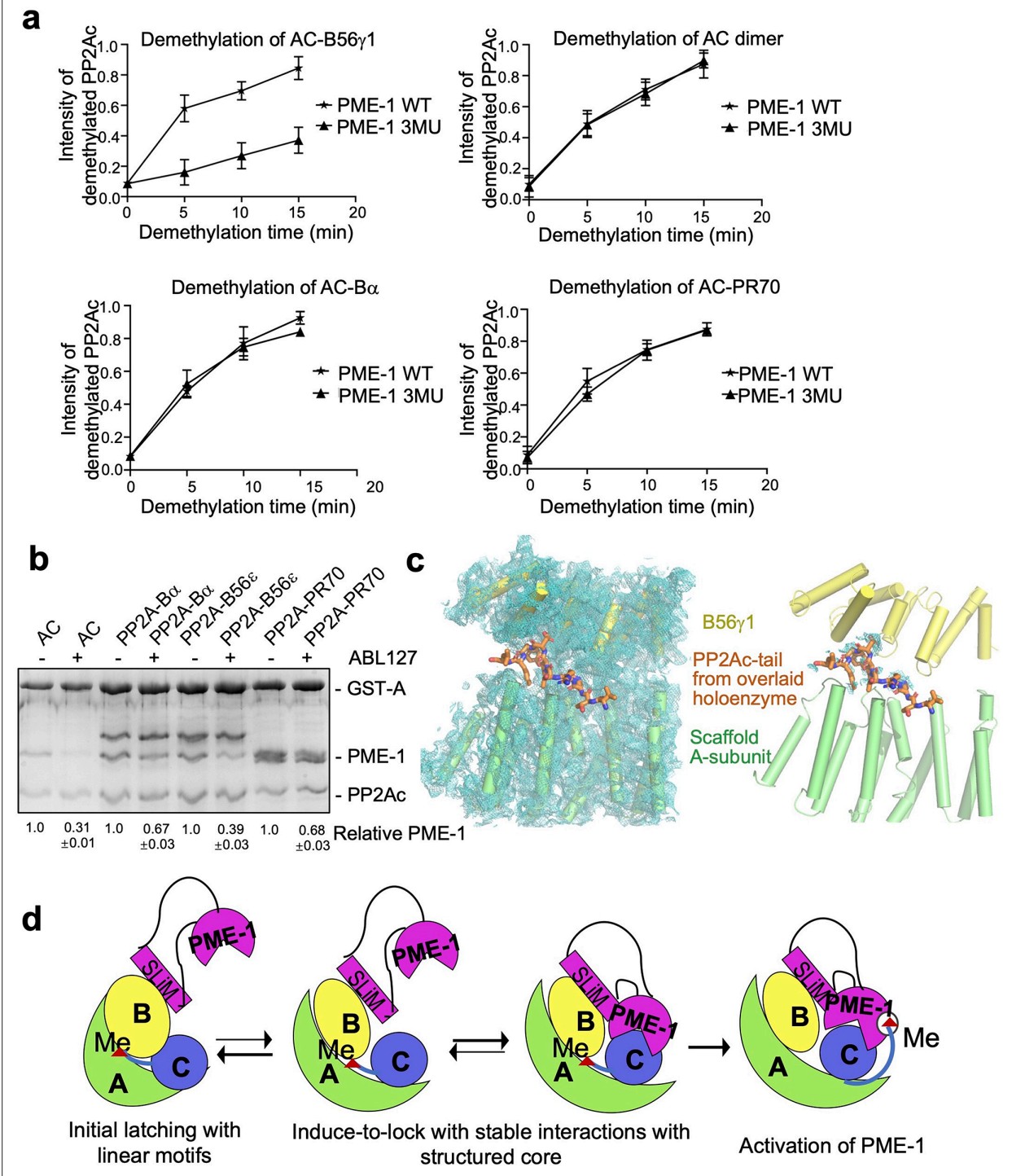

**Figure 5.** Different effects of PP2A methylesterase 1 (PME-1) inhibitor and B56–PME-1 interface mutations. (**a**) The time-dependent demethylation curves of protein phosphatase 2A (PP2A)-B56γ1 holoenzyme (upper left), the AC dimeric core enzyme (upper right), PP2A-Bα holoenzyme (lower left), and PP2A-PR70 holoenzyme (lower right) by PME-1 WT and mutant (3MU). The demethylation curves are plotted based on the quantified data in *Figure 5—figure supplement 1*. (b) Pulldown of PME-1 by GST-tagged PP2A core enzyme or PP2A-Bα/PP2A-B56ε/PP2A-PR70 holoenzymes in the presence or absence of PME-1 inhibitor, ABL127. Proteins bound to Glutathione-Sepharose 4B (GS4B) resins were examined and quantified similar to *Figure 4e*. The intensities of bound PP2A were first normalized to the immobilized GST-tagged proteins and then normalized to control (1.0). (**c**) Illustration of the A–B56γ1 interface and the PP2Ac tail from the overlaid holoenzyme. The cryoelectron microscopy (cryo-EM) map was shown at 3.5 Å. The electron density map for the A-subunit and B56γ1 was shown (left). The lack of cryo-EM electron density for the PP2Ac tail that occupies the A–B56 interface in the overlaid holoenzyme (right) supports the notion that the PP2Ac tail disappears from this interface once the holoenzyme is bound

*Figure 5 continued on next page*

*Figure 5 continued*

to PME-1. A-subunit, B56γ1, and electron density map are colored as in **Figures 3 and 4**. The PP2Ac tail from the holoenzyme is shown in stick and colored orange. (**d**) The latch-to-induce-and-lock model for PME-1 interactions with and methylesterase activation toward PP2A holoenzymes. In brief, PME-1 disordered motifs, including substrate-mimicking SLiM, latch to the regulatory subunit, enabling holoenzyme conformational changes, PME-1 interactions with PP2Ac, and the movement of PP2Ac tail from the holoenzyme interface to the PME-1 active site.

The online version of this article includes the following source data and figure supplement(s) for figure 5:

**Source data 1.** Source data for *Figure 5a*.

**Source data 2.** Source data for *Figure 5b*.

**Figure supplement 1.** Time course of protein phosphatase 2A (PP2A) holoenzyme demethylation.

**Figure supplement 1—source data 1.** Source data for *Figure 5—figure supplement 1a*.

**Figure supplement 1—source data 2.** Source data for *Figure 5—figure supplement 1b*.

**Figure supplement 1—source data 3.** Source data for *Figure 5—figure supplement 1c*.

**Figure supplement 1—source data 4.** Source data for *Figure 5—figure supplement 1d*.

**Figure supplement 2.** Structural model illustrating *B* factors of the cryoelectron microscopy (cryo-EM) structure of the protein phosphatase 2A (PP2A)-B56γ1–PP2A methylesterase 1 (PME-1) complex.

interface mutations allow us to pinpoint these PME-1 functions to its activity toward PP2A-B56 holo-enzymes at both basal conditions and during response to DNA damage (**Figure 6g, h**), suggesting a better strategy to target PME-1 than the active site inhibitor.

## Discussion

Our studies reveal structural malleability and functional versatility of PME-1 for both diverse PP2A holoenzymes and complex cellular signaling. Since our first in vitro observation on PME-1–PP2A holo-enzyme interactions (**Figure 1—figure supplement 1c**), it has been more than a decade to gain the current level of insights into the structural and biochemical mechanisms of PME-1 toward PP2A holoenzymes. This advance is greatly facilitated by accumulated knowledge on PP2A holoenzyme biogenesis, recycling, and substrate recognition (**Jiang et al., 2013**; **Guo et al., 2014**; **Stanevich et al., 2011**; **Wu et al., 2017b**; **Wlodarchak et al., 2013**; **Wu et al., 2017a**). The underlying structural and biochemical insights demonstrate significant diversification of PME-1 activities toward PP2A holo-enzymes and cellular signaling by coupling to substrate-mimicking SLiMs.

PME-1-mediated holoenzyme demethylation would remove the methylation mark that protects PP2A holoenzymes from disassembly by α4 and TIPRL (**Wu et al., 2017b**) and provides a mechanism to prime PP2A holoenzymes for disassembly (**Figure 7**). In vitro dissection of PP2A regulation complexes arrives at a strictly controlled linear pathway for PP2A holoenzyme biogenesis: (1) stabilization of the partially folded, latent PP2Ac by α4 (**Jiang et al., 2013**); (2) PP2Ac activation by phosphatase activator (PTPA) (**Guo et al., 2014**); (3) methylation of the active core enzyme by LCMT-1 (**Stanevich et al., 2011**); and (4) methylation-facilitated formation of stable holoenzymes in cells (**Bryant et al., 1999**; **Tolstykh et al., 2000**). Together with this pathway, holoenzyme demethylation and disassembly form a regulation loop for holoenzyme biogenesis and recycling (**Figure 7**), which provides a plausible mechanism for up- and downregulation of PP2A holoenzymes in cellular signaling. Periodic PP2A demethylation at different cellular locations occurs during cell cycle (**Turowski et al., 1995**), suggesting that PP2A holoenzymes might undergo cell cycle-dependent disassembly. PP2A recycling might also occur during stress or DDR, creating a window of reduced PP2A holoenzyme activity to propagate DDR signaling cascades (**Kong et al., 2009**). The ability to probe holoenzyme reshuffling and in-depth understanding of PP2A holoenzyme function and substrate recognition in cellular signaling would be crucial for investigating such dynamic holoenzyme homeostasis in cells, which remains critical gaps in PP2A biology.

The versatile PME-1 functions toward the PP2A core enzyme and diverse holoenzymes contrast the strictly controlled LCMT-1 activity toward the active core enzyme and likely explains its more restricted cellular level and cellular location than LCMT-1. While LCMT-1 is highly abundant in cells under normal conditions (**Stanevich et al., 2011**), elevated PME-1 level is associated with many types of cancer (**Kaur et al., 2016**; **Wandzioch et al., 2014**) and neurological disorders (**Nicholls et al., 2016**; **Ortega-Gutiérrez et al., 2008**; **Sontag et al., 2010**). Our study reveals detailed mechanisms

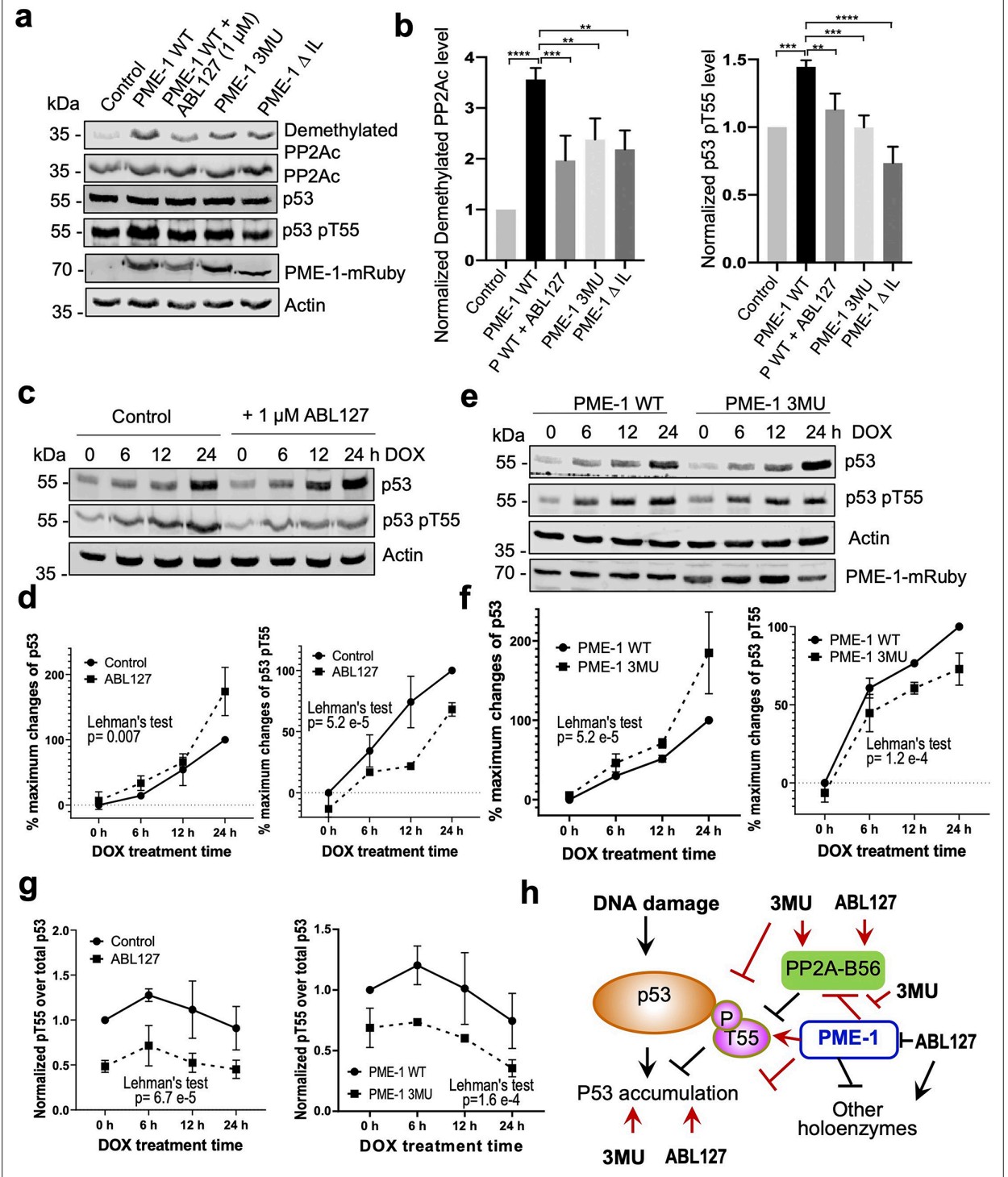

**Figure 6.** PP2A methylesterase 1 (PME-1) function in p53 signaling. (**a**) Effects of PME-1 inhibitor versus B56 interface mutations in PME-1 (3MU) on p53 phosphorylation at Thr55 and cellular PP2Ac methylation. The mock vector, wild-type, or mutant PME-1 were transfected into 293T cells, followed by treatment by ABL127 as indicated. Total cell lysates were examined by western blot. The representative results from three repeats were shown. (**b**) Cellular levels of p53 pThr55 and demethylated PP2Ac in (**a**) were normalized against actin loading control, and mean ± standard deviation (SD) was calculated from three independent experimental repeats. One-way analysis of variance (ANOVA) with Tukey's multiple comparisons was used to determine the difference between independent groups (**p < 0.01; ***p < 0.001; ****p < 0.0001). (**c**) Effects of PME-1 inhibitor on p53 accumulation and phosphorylation at T55 during DNA damage response (DDR). 293T cells were treated with DOX (2 μg/ml) in the presence of ABL127 (1 μM) or Dimethyl sulfoxide (DMSO) for indicated durations. Total p53 and pThr55 were detected at indicated time points. The representative results from three

*Figure 6 continued on next page*

*Figure 6 continued*

repeats were shown. (**d**) Time-dependent increases in p53 total protein and pThr55 were first quantified as the percentage of maximum changes in each experimental repeat in **c** and mean ± SD were then calculated from three repeats and used for data plotting. (**e**) Effects of PME-1 3MU mutations on p53 accumulation and phosphorylation at T55 during DDR. 293T Cells were treated with DOX (2 µg/ml) after 48 hr transfection with wild-type or mutant PME-1. Total p53 and pThr55 were detected at indicated time points. The representative results from three repeats were shown. (**f**) Time-dependent increases in p53 total protein and pThr55 were quantified as in **d** using data in e. (**g**) Ratios of pThr55 over total p53 during DDR calculated from data in **c** (left) and **e** (right). Mean ± SD were calculated from three independent experimental repeats. (**d–g**) The difference between curves was evaluated using Lehman's test. (**h**) Illustration of p53 signaling during DDR and roles of PME-1 in suppressing p53 accumulation by downregulating protein phosphatase 2A (PP2A)-B56 holoenzyme activities toward the inhibitory phosphorylation at Thr55 of p53. 3MU selectively abolishes the PME-1 activities toward the PP2A-B56 holoenzymes. Thus, its function in enhancing p53 accumulation and suppressing pThr55 in p53 is mediated by its specific function to alleviate PME-1 activities toward PP2A-B56 holoenzymes. ABL127 blocks total PME-1 activities toward all PP2A holoenzymes, including the PP2A-B56 holoenzymes, and thus affects total p53 and pThr55 similar to 3MU.

The online version of this article includes the following source data and figure supplement(s) for figure 6:

**Source data 1.** Source data for *Figure 6a*.

**Source data 2.** Source data for *Figure 6b*.

**Source data 3.** Source data for *Figure 6c*.

**Source data 4.** Source data for *Figure 6d*.

**Source data 5.** Source data for *Figure 6e*.

**Source data 6.** Source data for *Figure 6f*.

**Source data 7.** Source data for *Figure 6g*.

**Figure supplement 1.** Amplification and mutation of the *PPME-1* gene that encodes PP2A methylesterase 1 (PME-1) in cancer are associated with poorer survival outcome.

of elevated PME-1 expression in suppressing p53 activity both under basal conditions and during DDR (*Figure 6*), corroborating the oncogenic function of PME-1. Furthermore, PME-1 activity is spatially controlled in cells (*Longin et al., 2008*; *Turowski et al., 1995*). Robust PP2A demethylation was detected after the mammalian cells were lysed and predominantly affected PP2A-B55 and PP2A-PR72 holoenzymes (*Yabe et al., 2018*). These holoenzymes are likely spatially separated from PME-1 in intact cells but interact upon disruption of cellular organelles. Since PP2A regulatory subunits are

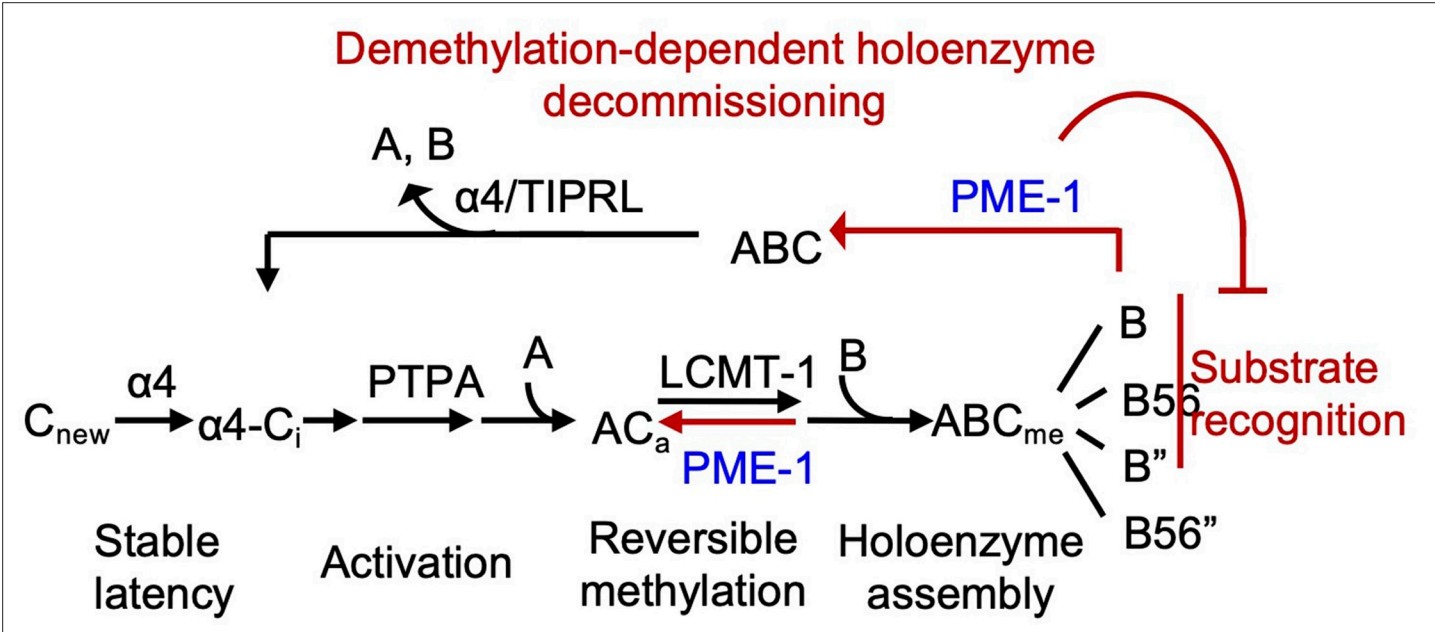

**Figure 7.** Illustration of the signaling loop of protein phosphatase 2A (PP2A) holoenzyme biogenesis and recycling and the multifaceted roles of PP2A methylesterase 1 (PME-1), including demethylation of the PP2A core enzyme to suppress holoenzyme assembly, inhibition of holoenzyme substrate recognitions, and demethylation of PP2A holoenzymes. The latter provides a mechanism for priming PP2A holoenzymes for demethylation-dependent decommissioning.

highly varied in tissue, organ, and cellular distributions, holoenzyme demethylation might occur in strictly and spatiotemporally controlled manners under distinct signaling contexts. How PME-1 activities are regulated and how its broad covalent modifications (http://phosphosite.org) regulate its activities remain to be investigated.

The cryo-EM structure of the PP2A-B56 holoenzyme–PME-1 complex reveals remarkable structural malleability of both the holoenzyme and PME-1 (*Figure 3*). This observation reinforces the dynamic nature of PP2A regulation, although holoenzymes are thermodynamically the most stable forms of PP2A. The structural dynamic not only accommodates the unexpected binding but is also crucial for methylesterase activation. The structured cores and disordered regions make at least six separate contacts between PME-1 and PP2A-B56 holoenzymes (*Figures 2–4*). These contacts involve multiple functionally important sites, including the PME-1 active site, the phosphatase active site, and the B56 substrate-binding groove, providing a coherent structural basis for holoenzyme demethylation and for suppressing PP2A holoenzyme activity. In summary, the mode of PME-1 binding to the PP2A-B56 holoenzyme underlies four coherent mechanisms to control the holoenzyme functions: (1) holoenzyme demethylation to prime holoenzyme for decommissioning; (2) reducing the efficacy of substrate-entry to the phosphatase active site by direct binding to the PP2A active site and by (3) perturbing the B56 internal loop near the active site (*Figure 3b, c* and *Figure 3—figure supplement 4*); (4) blocking the protein groove for the LxxIxE motif in substrates (*Figure 4a, b*). Reciprocally, the shared holoenzyme-binding motifs in PME-1 and substrates also render holoenzymes engaged with substrates less prone to demethylation. The PME-1 disordered regions synthesize novel activities by three mechanisms: tethering holoenzymes, enabling large structural shifts, and blocking holoenzyme–substrate recognition. Such structural and functional intricacy is likely a common theme for PME-1 interactions with other families of PP2A holoenzymes.

Our in vitro structural and biochemical insights provide a foundation for dissecting complex cellular PME-1 activities. Both holoenzyme–substrate and holoenzyme–PME-1 interactions are highly dynamic and flimsy in cells, making cellular investigation extremely challenging. The strict control of cellular location and activities of PME-1 also requires experimental strategies to dissect PME-1 functions in intact cells. The general effects of PME-1 inhibitor and the selective effects of B56-specific interface mutations allow us to pinpoint the function of PME-1 toward PP2A-B56 holoenzymes in controlling inhibitory p53 phosphorylation at Thr55 (*Figure 6*). Built on the previous observation that p53 pThr55 is a target site of PP2A-B56 holoenzymes (*Li et al., 2007*), we demonstrated a novel cellular PME-1 function in regulating pThr55 and total p53 both at basal conditions and during DDR via its activities toward PP2A-B56 holoenzymes (*Figure 6*).

SLiMs are extremely powerful in synthesizing regulation nodes and signaling networks, as demonstrated for the PPP family of phosphatases, such as SLiMs in calcineurin substrates (*Roy and Cyert, 2009*; *Wigington et al., 2020*) and PP4 (protein phosphatase 4) substrates (*Ueki et al., 2019*). Substrate SLiMs for PP2A-B56 holoenzymes and PP1 (protein phosphatase 1) orchestrate sequential events regulated by PP2A holoenzymes, PP1, and kinases for precise control of cell cycle progression (*Bollen, 2015*; *Hendrickx et al., 2009*; *Lesage et al., 2011*). Our studies demonstrate that the combination of dynamic sets of structured cores and SLiMs creates more versatile activities in cellular signaling. Such combination might be a common theme for PP2A and other cellular signaling complexes. Efforts along this line would build our ability to overcome major challenges in deciphering dynamic PP2A functions and regulations in broad cellular processes.

## Materials and methods
### Protein preparation

All protein constructs were generated by standard PCR molecular cloning strategy. PME-1, B56γ1, and their mutants were cloned into pQlink vector (Addgene, Cambridge, MA, USA), and proteins were overexpressed in *Escherichia coli* DH5α at 23°C overnight. The PP2A Aα, B56ε, PR70 (122–530), and CFP-Aα (9-589)-TC were overexpressed in *E. coli* DH5α at 23°C overnight (*Wlodarchak et al., 2013*; *Wlodarchak et al., 2013*; *Xing et al., 2006*). LCMT-1 was overexpressed at 37°C for 4 hr in *E. coli* BL21(DE3) (*Stanevich et al., 2011*). PP2Ac and B55α were overexpressed in Hi5 insect cells using Bac-to-Bac baculovirus expression system (*Stanevich et al., 2011*; *Zhang et al., 2018*; *Zhang et al., 2018*). The supernatant of cell lysate was purified over Glutathione-Sepharose 4B (GS4B) (GE

Healthcare, Boston, MA, USA) or Ni-NTA resins (Qiagen). After affinity purification, all tags were cleaved by TEV, thrombin, or PreScission protease. Proteins were further purified by ion exchange chromatography (Source 15Q, GE Healthcare) and gel filtration chromatography (Superdex 200, GE Healthcare). GST-tagged SYT16 peptide ([132]KLPHVLSSIAEEEHH[147]L), GST-CRTC3 peptides (290–401, 370–401, and 380–401 [380]SGPSRRRQPPVSPLTLSPGPE[401]A), and GST-Cdc6 peptide ([49]KALPLSPRKRLG DDNLCNTPHLPPCSPPKQGK KENGPPHSH[90]T) were cloned to pQlink vector, overexpressed in *E. coli* DH5α at 23°C overnight, and purified over GS4B resin and ion exchange chromatography. PP2A core enzyme was assembled as previously described (*Wlodarchak et al., 2013*; *Xing et al., 2006*).

## In vitro characterization and biochemical assays

FRET assay, methylation, methylesterase activity assay, comigration over gel filtration chromatography, ITC, GST-mediated pulldown assay and binding competition, and proteomic peptide phage display (Pro-PD) were described in supplemental materials.

## Cryo-EM sample preparation and data acquisition

The PP2A core enzyme was assembled as described (*Xing et al., 2006*) and then methylated by LCMT-1 in the presence of *S*-adenosyl methionine (SAM). The methylated PP2A core enzyme was incubated with excess amounts of B56γ1 and PME-1 (1.1- and 1.2-fold over PP2A core enzyme, respectively) containing an inactive mutation (S156A). The PP2A–PME-1 complex was purified to homogeneity by gel filtration chromatography. Purified PP2A–PME-1 complex was crosslinked by incubating with 0.05% glutaraldehyde for 15 min at RT and then quenched with 0.1 M Tris (pH 8.0). The crosslinked PP2A–PME-1 complex was further purified by gel filtration chromatography and concentrated to 1 mg/ml. For cryo-sample preparation, 3 µl of purified PP2A–PME-1 complex was applied onto a glow-discharged holey carbon grid (Quantifoil 300 mesh R 1.2/1/3 with ultrathin carbon). Grid was immediately blotted for 4 s with a blot force of 0 and plunge frozen in liquid ethane using Vitrobot (Thermo Fisher Scientific) at 4°C and 100% humidity. Cryo-EM data were collected using a Titan Krios operating at 300 kV with a Gatan K3 detector and GIF Quantum energy filter. Movie stacks were collected using SerialEM, with a slit width of 20 eV on the energy filter and a defocus range from −1.5 to −2.3 µm in superresolution counting mode at a magnification of ×81,000, corresponding to a physical pixel size of 1.059. Each stack was exposed for 3.2 s with an exposure time of 0.05 s per frame, resulting in 64 frames per stack. The total dose rate was 50.8 e−/Å² for each stack.

## Cryo-EM data processing

Movie frames were aligned using the Motioncorr2 (*Zheng et al., 2017*). The contrast transfer function (CTF) parameters were estimated from the aligned micrographs using CTFFIND4 (*Rohou and Grigorieff, 2015*). Automated particle picking first using 1000 images, particle extraction with a box size of 480 pixels, and two-dimensional (2D) classification were performed in cryoSPARCv2.14 (*Punjani et al., 2017*). High-quality 2D class averages representing projections in different orientations were selected as templates for automatic particle picking of the entire dataset. Three rounds of 2D classification yielded 801,656 particle images with clear features of the PP2A–PME-1 complex. After ab initio model building, 3D classification into three classes with two reiterations to remove bad particles was performed using cryoSPARCv2.14 heterogeneous refinement, followed by homogenous refinement for the best class. Local refinement yielded an improved map with better details at a resolution of 3.4 Å using cryoSPARCv3.2. The resolution was estimated by applying a soft mask around the protein complex and using the gold-standard Fourier shell correlation (FSC) = 0.143 criterion.

## Model building and refinement

The initial model of the PP2A–PME-1 complex was built in Pymol based on the structure of PP2A-B56γ1 holoenzyme (PDB code: 2NPP) and PME-1–PP2A core enzyme (PDB code: 3C5W), and manually docked into the 3.4 Å map in Chimera v1.15 and adjusted in COOT v0.9.6 (*Emsley et al., 2010*). The B56 SLiM in the PME-1 internal loop (residues [251]VEGIIE[257]E) was modeled to the substrate-binding groove of B56γ1, guided by the trace of BubR1 B56 SLiM ([669]LDPIIE[675]D) in the B56γ1–BubR1 complex (PDB code: 5JJA). The structural model was refined using the phenix.real_space_refine program in PHENIX (v1.19.2) (*Adams et al., 2010*) with secondary structure and geometry restraints. The model was analyzed using MolProbity (*Chen et al., 2010*).

## Mammalian cell culture and co-IP

Human embryonic kidney cells (HEK293T, ATCC, Cat#: CRL-3216) were cultured in Dulbecco's modified Eagle's medium (Gibco, Thermo Fisher Scientific, Waltham, MA, USA) with 10% fetal bovine serum (Hyclone, GE Healthcare, Boston, MA, USA), 100 U/ml penicillin, and 100 µg/ml streptomycin in a humidified atmosphere at 37°C with 5% $CO_2$. HEK293T cells were purchased from ATCC and have been certified free from mycoplasma contamination and genetic authenticity, which was confirmed by PCR screen of mycoplasma.

The HA-tagged human B56 subunits and mRubby-tagged human PME-1 were cloned into murine retroviral vectors bearing a cytomegalovirus promoter. After cotransfection into 293T cells, the expression levels of recombinant B56 subunits and PME-1 were monitored by western blot using antibodies that specifically recognize HA-tag (Sigma, 2CA5, Cat#: 116666060011, 1:1000) and PME-1 (Abcam, Cat#: ab86409, 1:1000). The interaction between B56 subunits and PME-1 was detected by co-IP using anti-HA antibody immobilized on protein G magnetic beads (Invitrogen) to immunoprecipitate PME-1 bound to HA-B56 48 hr after transfection. Cells were lysed in lysis buffer (50 mM Tris–HCl pH 8.0, 150 mM NaCl, 1 mM ethylenediaminetetraacetic acid (EDTA), 1 mM dithiothreitol, and 0.5% Triton X-100), and 500 µg of cell extracts were immunoprecipitated at 4°C in lysis buffer for 2 hr followed by western blot. 50 µg whole-cell extracts were examined by western blot to assess the levels of recombinant protein expression. PP2Ac and Actin protein levels were detected by anti-PP2Ac antibody (Cell Signaling Technology, Cat#: 2259, 1:1000) and anti-β-Actin antibody (Cell Signaling Technology, Cat#: 3700, 1:1000), respectively. The experiments were repeated three times, and the representative results were shown.

## Immunoblotting

HEK293T cells were transfected with WT or mutant PME-1 expression vectors using Lipofectamine 2000 (Thermo Fisher, Cat#: 11668019), and 48 hr after transfection, the cells were treated with doxorubicin (Dox) (2 µg/ml) to induce DNA damage for the indicated amounts of time. Treatments of HEK293T cells with Dox (2 µg/ml) and/or ABL127 (1 µM) or DMSO control were carried out after the cell confluency reached 75%. At the indicated time of treatments, cells were collected and suspended in ice-cold RIPA lysis buffer containing 150 mM NaCl, 0.1% Triton X-100, 0.5% sodium deoxycholate, 0.1% sodium dodecyl sulfate, 50 mM Tris–HCl pH 8.0, protease inhibitor cocktail (Sigma-Aldrich, Cat#: 11836170001), and phosphatase inhibitor cocktail (Sigma-Aldrich, Cat#: 4906845001). The lysates were centrifuged at 13,000 × $g$ for 20 min at 4°C, and the supernatant was collected. The whole protein (30 µg) was analyzed by western blot. The total p53 and its phosphorylation at Thr55 were detected by anti-p53 antibody (Cell Signaling Technology, 1C12, Cat#: 2524, 1:1000) and anti-p53 pThr55 antibody (Abcam, Cat#: ab183546, 1:1000). Actin level was detected by anti-β-Actin antibody (Cell Signaling Technology, Cat#: 3700, 1:1000) as loading control. The experiments were repeated three times, and the representative results were shown. Data analysis and statistical studies were performed in GraphPad (Prism Inc) and Mstat 7.0 (oncology.wisc.edu/mstat).

## Acknowledgements

We thank Drs Janette Myers and staff at the Pacific Northwest Center for Cryo-EM (PNCC) at Oregon Health & Science University for assistance and discussions on cryo-EM data collection and processing, Kenneth Satyshur and Ivan Andrewjeski from the School of Medicine of UW-Madison for IT support, and Dr. Norman Drinkwater from our department for consulting on statistical analysis. We thank Drs Robert Kirchdoerfer and Elizabeth Wright from UW-Madison and Ning Yan from Princeton University for discussions on cryo-EM studies. We thank Dr. Paul Lambert from Oncology Department of UW-Madison for reading the manuscript. The work is supported by American Cancer Society Research Scholar Grant RSG-10-153-01-DMC (YX), National Institute for General Medicine R01s GM096060-01, GM137090-01 (YX), Jordan's Guardian Angels Foundation grant and Jordan's Syndrome research consortium fund from the State of California A19-3376-5007 (YX).

## Additional information

### Funding

| Funder | Grant reference number | Author |
|---|---|---|
| National Institute of General Medical Sciences | GM137090-01 | Yongna Xing |
| American Cancer Society | RSG-10-153-01-DMC | Yongna Xing |
| Jordan's Guardian Angels Foundation and Jordan's Syndrome research consortium fund from the State of California | A19-3376-5007 | Yongna Xing |
| National Institute of General Medical Sciences | GM096060-01 | Yongna Xing |

The funders had no role in study design, data collection, and interpretation, or the decision to submit the work for publication.

### Author contributions

Yitong Li, Conceptualization, Resources, Data curation, Formal analysis, Validation, Investigation, Methodology, Project administration, Writing – review and editing; Vijaya Kumar Balakrishnan, Data curation, Formal analysis, Validation, Investigation, Methodology; Michael Rowse, Ylva Ivarsson, Conceptualization, Data curation, Formal analysis, Investigation, Methodology; Cheng-Guo Wu, Data curation, Validation, Investigation, Methodology; Anastasia Phoebe Bravos, Data curation, Investigation, Methodology; Vikash K Yadav, Data curation, Formal analysis, Investigation, Methodology; Stefan Strack, Resources; Irina V Novikova, Data curation, Formal analysis; Yongna Xing, Conceptualization, Resources, Data curation, Formal analysis, Supervision, Funding acquisition, Validation, Investigation, Methodology, Writing - original draft, Project administration, Writing – review and editing

### Author ORCIDs

Yitong Li ⓘ http://orcid.org/0000-0002-7866-8460
Michael Rowse ⓘ http://orcid.org/0000-0002-5572-5258
Yongna Xing ⓘ http://orcid.org/0000-0002-9834-528X

### Decision letter and Author response

Decision letter https://doi.org/10.7554/eLife.79736.sa1
Author response https://doi.org/10.7554/eLife.79736.sa2

## Additional files

### Supplementary files
• MDAR checklist

### Data availability

All data are available in the main text or in the supplementary materials. The cryo-EM map and the refined atomic model of PP2A-B56γ1–PME-1 complex have been deposited at EMDB and RCSB under the accession codes of EMD-25363 and 7SOY, respectively.

The following datasets were generated:

| Author(s) | Year | Dataset title | Dataset URL | Database and Identifier |
|---|---|---|---|---|
| Li Y, Balakrishnan VK, Novikova IV, Xing Y | 2022 | The structure of the PP2A-B56γ1 holoenzyme-PME-1 complex | https://www.ebi.ac.uk/emdb/EMD-25363 | Electron Microscopy Data Bank, EMD-25363 |
| Li K, Balakrishnan BK, Novikova IV, Xing Y | 2022 | The structure of the PP2A-B56γ1 holoenzyme-PME-1 complex | https://www.rcsb.org/structure/7SOY | RCSB Protein Data Bank, 7SOY |

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

## Appendix 1

### Supplemental materials and methods

#### FRET assay

FRET assay was performed as described previously (*Wlodarchak et al., 2013*). The fluorescent donor signals of CFP of three PP2A holoenzymes containing CFP-Aα (9-589)-TC fusion protein (100 μg/ml) were measured with excitation at 450 nm and emission at 490 nm in the presence or absence of FLAsH-EDT$_2$ compound (Invitrogen, Cat#: T34561) using ClarioStar Plate Reader (BMG LABTECH). FLAsH-EDT$_2$ compound (1.1-fold) was conjugated to CFP-Aα (9-589)-TC (1-fold) to generate the TC-FLAsH fluorescent group, which acts as an acceptor in the FRET assay. The rate of energy transfer was calculated based on the loss of donor fluorescence using the following equation: $E = 1 - (F_{DA}/F_D)$, where $F_{DA}$ and $F_D$ are the fluorescence of CFP in the presence and absence of TC-FLAsH, respectively. The rate of energy transfer was recorded before and after adding a fivefold molar amount of PME-1.

#### Methylation and demethylation assay

Methylation assays were performed as previously described (*Stanevich et al., 2011*; *Xing et al., 2006*). PP2A core enzyme (1 μM) was mixed with equal molar amounts of LCMT-1 and PTPA in the presence of *S*-(5'-adenosyl)-L-methionine (SAM) in the reaction buffer containing 25 mM Tris pH 8.0, 150 mM NaCl, 10 mM dithiothreitol (DTT), 50 μM MnCl$_2$, and 10 mM ascorbic acid. The reaction mixture was incubated at 4°C overnight. Methylated PP2A core enzyme was purified by ion exchange chromatography (Source 15Q, GE Healthcare) and used to assemble different holoenzymes. Methylated AC dimer or PP2A holoenzymes (100 nM) and PME-1 (20 nM) at a 1:0.2 molar ratio were incubated at 30°C for 15 min. The demethylation level was measured by dot blot or western blot using an antibody that specifically recognizes the unmethylated PP2Ac (Millipore, Cat#: 05-5774b7, 1:1000). Total PP2Ac was assessed after the methyl group was completely removed by treatment with 100 mM NaOH. Samples containing 8 ng of PP2Ac were spotted for dot blot, and 16 ng were applied for western blot. Anti-PP2A A-subunit antibody (Cell Signaling Technology, Cat#2039, 1:1000) was used to detect A-subunit in the western blot as the loading control.

#### Comigration over gel filtration chromatography

0.3 mg of PP2A core enzyme was mixed with 1.1-fold B subunit (Bα, B56γ1, B56δ, B56ε, or PR70) and 1.5-fold PME-1 (FL, ΔN18, ΔIL) and subjected to gel filtration chromatography (Superdex 200, GE Healthcare). The protein fractions were analyzed by SDS–PAGE and visualized by Coomassie blue staining.

#### Isothermal titration calorimetry

Binding affinities were measured by ITC. Briefly, 20 μM PME-1 (FL, ΔN18, or ΔIL) was titrated with 300 μM B56γ1 or PR70 with a Nano-ITC (TA instruments) at 22°C. For B56 SLiM binding to B56γ1, 20–40 μM B56γ1 was titrated with 200–300 μM GST-PME-1 246–263 (GSKSI$^{251}$VEGIIE$^{257}$EEEEDEEG) or GST-SYT16 peptide (LSSIAEEE) with a Nano-ITC (TA instruments) at 15°C, a lower temperature to avoid B56γ1 precipitation in the cell. All proteins were prepared in a buffer containing 25 mM N-2-hydroxyethylpiperazine-N'-2-ethanesulfonic acid (HEPES) (pH 7.5) and 150 mM NaCl.

#### GST-mediated competitive binding assay and pulldown assay

Approximately 30 μg of GST-SYT16, GST-CRTC3, or GST-Cdc6 was immobilized to 5 μl of glutathione magnetic agarose beads (Thermo Scientific) via GST tag. The beads were washed with 200 μl binding buffer containing 25 mM Tris (pH 8.0), 150 mM NaCl, and 0.1% Triton-100 three times to remove the unbound protein. Different concentrations of PME-1 (0, 20, 40, and 60 μM) or truncated mutants were mixed with 20 μM PP2A holoenzymes and then added to the beads in 100 μl binding buffer. The mixture was incubated for 20 min at room temperature and washed three times with the binding buffer. The proteins bound to beads were analyzed by SDS–PAGE and visualized by Coomassie blue staining.

To test the interaction between Bα and PME-1 FL, ΔIL, or ΔN18, 20 μg His-tagged Bα was immobilized to 5 μl Ni-NTA magnetic agarose beads (Thermo Scientific). The beads were washed with 200 μl binding buffer three times to remove the unbound protein. 0.25, 1, or 4 μM PME-1 FL, ΔIL, or ΔN18 in 100 μl binding buffer was added to the beads and incubated for 30 min at room

temperature, respectively. The beads were washed three times, and the proteins bound to the beads were analyzed by SDS–PAGE and visualized by Coomassie blue staining.

To examine the interaction between B56γ1 (or the PP2A-B56ε holoenzyme) and WT or mutant PME-1, 30 µg GST-B56γ1 (or GST- PP2A-B56ε holoenzyme) was immobilized to the magnetic glutathione beads. The beads were washed with 200 µl binding buffer three times to remove the unbound protein. 20 µM PME-1 WT or mutants in 100 µl binding buffer was added to the beads and incubated for 20 min at room temperature. The beads were washed three times, and the proteins bound to the beads were analyzed by SDS–PAGE and visualized by Coomassie blue staining.

A similar method was used to test the effect of ABL127 on the interaction between PME-1 and PP2A. In brief, 10 µM WT PME-1 was preincubated with 50 µM ABL127 diluted 20 times from 1 mM stock or 5% DMSO control for 15 min and then mixed with glutathione resin with immobilized GST-tagged PP2A core enzyme or PP2A-B56ε holoenzyme (30 µg), followed by the pulldown procedure described above. All experiments were repeated three times.

## Proteomic peptide phage display

The PP2A-B56γ1 holoenzyme and its complex with PME-1 were used as bait proteins in selection against a phage library that displays 16-residue tiled peptides from disordered regions of the human proteome following the same procedure as previously described (*Wu et al., 2017a*). In brief, GST-tagged reconstituted complexes (25 µg) or GST (20 µg) were allowed to associate with GSH-conjugated magnetic beads (20 µl, 1:1 bead/buffer slurry; Thermo Fisher Scientific, Waltham, MA, USA) for 2 hr, under gentle shaking at 4°C. The beads were pelleted using a magnetic stand, and the supernatant was removed. Before biopanning, the beads were washed four times with 1 ml TBS. Four successive rounds of phage selection and amplification were performed. Bound phages were eluted by incubation with 100 µl of 100 mM HCl for 5 min at RT with gentle shaking. The acid-eluted phage pools were neutralized by adding 15 ml of 1.0 M Tris–HCl, pH 11.0, and used to infect *E. coli* Omnimax for amplification before the following selection rounds.

Phage pools were barcoded for next-generation sequencing. Undiluted amplified phage pools (5 µl) were used as templates for 24 cycles 50 µl PCR reactions using a distinct set of barcoded primers (0.5 µM each primer) for each reaction, and Phusion High Fidelity DNA polymerase (NEB) with a maximum polymerase concentration. PCR reactions were supplemented with Gel Loading Dye Purple (6×) (NEB) and separated on a 2.5% low melt agarose (BioRad, Hercules, CA, USA) gel stained with Roti-Safe GelStain (Carl-Roth, Karlsruhe, Germany). The DNA was visualized by UV light. The PCR products were extracted using the QIAquick Gel Extraction Kit (Qiagen, Hilden, Germany) according to the manufacturer with the following exceptions: (1) Gel extracts were resolved at room temperature (RT); (2) DNA was eluted with 30 µl low Tris–EDTA (TE) buffer (Thermo Fisher Scientific). Molarities of the eluted library DNA were determined on the 2100 Bioanalyzer using the High Sensitivity DNA Kit (Agilent, Santa Clara, CA, USA). Template preparation was performed according to the manufacturer's instruction using the Ion PGM Template OT2 200 Kit on the Ion OneTouch 2 System (Thermo Fisher Scientific). Twenty-five µl of 5 pM library DNA ($1.25 \times 10^{-4}$ pmol) was used in the template reaction. Sequencing was conducted on the Ion Torrent PGM sequencer using the Ion PGM Sequencing 200 Kit v2 and the Ion 314 Chip v2 (Thermo Fisher Scientific) according to the manuals. Signal processing and base calling were done using the Torrent Suite Software (Thermo Fisher Scientific).

