## [Editor Report]

Xing and colleagues used a combination of biochemical assays and cryo-EM to investigate the role of PME-1 in regulating PP2A, which plays an important role in tumorigenesis. Notably, they reveal here that PME-1 inserts an unstructured loop into the B-domain of the PPA2 holoenzyme and allosterically regulates the activity of the catalytic domain. This novel mechanism then plays a key role in controlling the cellular homeostasis of PP2A. Together, the work presented here provides new insights into mechanisms for an oncogenic function of PME-1 in regulating (inhibiting) p53 phosphorylation via PP2A-B56 holoenzymes under normal and DNA damage response conditions.

---

## [Decision Letter]

**Decision letter after peer review:**

Thank you for submitting your article "Coupling to short linear motifs creates versatile PME-1 activities in PP2A holoenzyme demethylation and inhibition" for consideration by *eLife*. Your article has been reviewed by 3 peer reviewers, one of whom is a member of our Board of Reviewing Editors, and the evaluation has been overseen by Volker Dötsch as the Senior Editor. The following individual involved in review of your submission has agreed to reveal their identity: Amy Whitaker (Reviewer #2).

Essential revisions:

Please address the remaining concerns of the reviewers.

*Reviewer #1 (Recommendations for the authors):*

– The use of red and green as colors for the two A-subunit structures that are being compared in Figure 3b could be chosen more wisely in case of red-green colorblind readers. It is a minor change but switching to a color pair like light green and dark green could help given the important points the authors make about the conformational changes in the A-subunit.

– The use of "left" and "right" panels in Figure 3 are somewhat difficult to track as a reader, especially since the structures in the bottom right area of Figure 3b bleed into the top of Figure 3c. A clear visual distinction should be made between which structures belong to 3b and 3c. This could be remedied by shrinking the right panel of 3c and shifting it up, or for even more clarity, the right panels could be broken down further into Figure 3d, 3e, etc.

*Reviewer #2 (Recommendations for the authors):*

This reviewer has no major concerns about the manuscript and isn't requesting any major revisions.

1. Page 10, line 209: There appears to be some words missing from the sentence making it unclear.

2. The methods contained a handful of typos and should be read carefully for accuracy.

*Reviewer #3 (Recommendations for the authors):*

The authors should address the following points prior to publication.

1) Page 3, line 45: for variable B subunits, should not use punctuation marks, instead, should use prime, double prime, and triple prime symbols.

2) Page 4, line 61: the PP2Ac tail sequence should be "TPDYFL" but not "RPDYFL".

3) Figure 1b and Figure S1b-d: please add Y-axis and molecular weight standards.

4) Figure 1d: what is the band marked by asterisk. Please clarify.

5) Figure 2b-d: the presentation of the SEC data and SDS-PAGE gels needs improvement:

– Please add MW standards.

– All SEC profiles should be in the same elution volume range of 6-18ml for better comparison.

– It's not clear why gels have different numbers of lanes, such as 7 to 9? The gel quality can be improved.

– Figure 2d right panel: the elution volume doesn't make sense. Why ACBα eluted at 14ml instead of 12ml?

6) The authors should provide an explanation of why ∆N18 deletion of PME-1 is used in their study.

7) Page 17, line 345: '…similar to the substrate peptide from BubR1', please add the reference and BubR1 substrate sequence.

8) Page 20, line 401: 'PME-1- 'should be 'PME-1.'

9) Figure 5d: for the latch-to-induce-and-lock model, the authors should comment on the possible effect on PME-1 binding to PP2A if changing the length of the linker between SLiM and PME-1 catalytic domain.

---

## [Author Response]

Reviewer #1 (Recommendations for the authors):– The use of red and green as colors for the two A-subunit structures that are being compared in Figure 3b could be chosen more wisely in case of red-green colorblind readers. It is a minor change but switching to a color pair like light green and dark green could help given the important points the authors make about the conformational changes in the A-subunit.

We greatly appreciate the comment. We changed the color following this reviewer’s suggestion. The red color was replaced by light color, helium, using Pymol.

– The use of "left" and "right" panels in Figure 3 are somewhat difficult to track as a reader, especially since the structures in the bottom right area of Figure 3b bleed into the top of Figure 3c. A clear visual distinction should be made between which structures belong to 3b and 3c. This could be remedied by shrinking the right panel of 3c and shifting it up, or for even more clarity, the right panels could be broken down further into Figure 3d, 3e, etc.

We greatly appreciate the comment. We divided the original Figure 3b – 3d into Figure 3b – 3f. The sizes and positions of the figure panels were also adjusted, as the reviewer suggested, to reflect their relationship more explicitly.

Reviewer #2 (Recommendations for the authors):This reviewer has no major concerns about the manuscript and isn't requesting any major revisions.1. Page 10, line 209: There appears to be some words missing from the sentence making it unclear.

We greatly appreciate that this reviewer identified the error. Please find our changes on lines 217-218 in the revised manuscript “…the interaction between the SYT16 peptide and PP2A-B56γ1 was blocked by…”.

2. The methods contained a handful of typos and should be read carefully for accuracy.

We greatly appreciate the comment. We corrected typos as much as possible.

Reviewer #3 (Recommendations for the authors):The authors should address the following points prior to publication.1) Page 3, line 45: for variable B subunits, should not use punctuation marks, instead, should use prime, double prime, and triple prime symbols.

Thanks for the comment. The changes were made at line 45 as suggested.

2) Page 4, line 61: the PP2Ac tail sequence should be "TPDYFL" but not "RPDYFL".

We greatly appreciate that the reviewer pointed out the error. It was corrected in the revised manuscript.

3) Figure 1b and Figure S1b-d: please add Y-axis and molecular weight standards.

Thanks for the comment. We added the Y-axis to all SEC profiles in Figure 1b and Figure S1b-d. We often use a variety of complexes and proteins with different molecular weights in the lab to normalize our size exclusion columns. In response to this reviewer’s comment, we added the peak positions of the PP2A free catalytic subunit (36 kDa), the PP2A core enzyme (100 kDa), and the PP2A-B56γ1 holoenzyme (150 kDa) as one set of lab standards to Figure 1b. The description of these standards was added to the figure legend for Figure 1b. We also marked the peak positions and molecular weights of the PP2A- B56γ1 holoenzyme, PME-1 alone, and the PME-1-holoenzyme complex in Figure S1b-d to demonstrate further the relationships between elution volumes and different molecular weights of the proteins/complexes.

4) Figure 1d: what is the band marked by asterisk. Please clarify.

Thanks for the comment. The bands marked by the asterisk were the heavy chain of the anti-HA antibody. We apologize for missing this point in the initial submission. The anti-HA antibody was immobilized on the Protein G beads and used to pull out the HA-tagged B56 and its associated proteins. It was then eluted together with the target proteins. The clarification was added to the figure legend of Figure 1d as suggested by this reviewer.

5) Figure 2b-d: the presentation of the SEC data and SDS-PAGE gels needs improvement:– Please add MW standards.

The MW standards were added as in Figure 1b and described in the figure legend.

– All SEC profiles should be in the same elution volume range of 6-18ml for better comparison.

Great suggestion. The SEC profiles in Figure 2b-d were revised as the reviewer suggested.

– It's not clear why gels have different numbers of lanes, such as 7 to 9? The gel quality can be improved.

The reviewer hinted at an important point that using the same number of lanes for the fractions in the same volume range would aid better comparison. The number of lanes varied because the peak fractions varied from profile to profile due to different binding affinities of the full-length or truncated PME-1 to different holoenzymes. We further cropped the SDS-PAGE image to keep the main fractions for the peak. In response to this reviewer's comment, we showed eight lanes for all profiles except Figure 2d right panel, which used nine lanes to reflect better the dissociation of PME-1 ∆N18 from the ACBα holoenzyme. Due to the tendency of free Bα to form aggregates, Bα is slightly sub-stoichiometric to the core enzyme. Therefore, a fraction of the PP2A core enzyme is associated with PME-1 ∆N18, and an additional lane was needed to show the excess amount of PME-1. Furthermore, as shown in Figure s2c, PME-1 ∆N18 reduces, rather than abolishes, the PME-1 binding to the ACBα holoenzyme. Therefore, a small amount of ACBα-PME-1 ∆N18 was still observed in the profile, resulting in a broader peak range.

As much as we attempted to present the fractions from the same range of elution volumes, there are still slight differences from profile to profile. Therefore, we also labeled the range of elution volumes for the fractions examined on SDS-PAGE for better clarity.

The image quality was in part compromised by the large number of PP2A complexes handled in parallel. The assembly of merely one PP2A holoenzyme complex is challenging for the majority of trainees. Handling a large number of complexes in parallel requires absolutely tenacious efforts. Even with a large batch of purified PP2Ac, after all the assembly and purification processes, the amounts of materials recovered for each complex became quite small. Therefore, a much larger volume for each fraction was boiled to a reduced volume for loading. We hope the reviewer recognizes the efforts behind those data and finds that the SDS-PAGE data are conclusive for whether PME-1 comigrates with the holoenzymes.

– Figure 2d right panel: the elution volume doesn't make sense. Why ACBα eluted at 14ml instead of 12ml?

We greatly appreciate that the reviewer identified this problem. The student used a different SEC column for this complex. Based on the MW standards, this profile is now normalized to the main column used for all other profiles.

6) The authors should provide an explanation of why ∆N18 deletion of PME-1 is used in their study.

In response to this reviewer’s comment, we added a brief clarification in the revised manuscript, lines 184-187:

“We observed that a complete deletion of the N-terminal disordered region or deletion of N-terminal residues beyond residue 18 tends to cause PME-1 aggregate or a poorer protein behavior. PME-1 ΔN18 was thus used to test the function of the N-terminal disordered region.”

7) Page 17, line 345: '…similar to the substrate peptide from BubR1', please add the reference and BubR1 substrate sequence.

The requested information was added to the revised manuscript, line 358: “similar to the substrate peptide from BubR1 (^669^LDPIIE^675^D) (J. Wang et al., 2016).”

8) Page 20, line 401: 'PME-1- 'should be 'PME-1.'

We apologize for the break of the phrase “PME-1-holoenzyme interactions” by the insertion of Figure 5. Figure 5 was inserted after the entire paragraph in the revised manuscript. This phrase is now in line 413.

9) Figure 5d: for the latch-to-induce-and-lock model, the authors should comment on the possible effect on PME-1 binding to PP2A if changing the length of the linker between SLiM and PME-1 catalytic domain.

The reviewer made a very interesting point. We elaborated on this point in the revised manuscript, lines 453-459:

“It is important to mention that the length of the PME-1 internal loop (239-282) is crucial in this “latch-to-induce-and-lock” model for the B56 SLiM (^251^VEGIIE^257^E) within the loop to reach the substrate SLiM-binding pocket of B56. As reflected in Figure 3a, the internal loop provides two invisible 12-residue and 25-residue linkers, 239-250 and 258-282, to span the direct distances of 23 Å and 31 Å, respectively, from the enzyme core to the two termini of the SLiM. Deleting any residues in the first linker might compromise the “latch-to-induce-and-lock” model.”

The addition of this structural analysis further supports the model.